# Determining the scale at which variation in a single gene changes population yields

**Erica McGale, Henrique Valim, Deepika Mittal[†], Jesús Morales Jimenez[‡], Rayko Halitschke, Meredith C Schuman[§], Ian T Baldwin\***

Department of Molecular Ecology, Max Planck for Chemical Ecology, Jena, Germany

**Abstract** Plant trait diversity is known to influence population yield, but the scale at which this happens remains unknown: divergent individuals might change yields of immediate neighbors (neighbor scale) or of plants across a population (population scale). We use *Nicotiana attenuata* plants silenced in mitogen-activated protein kinase 4 (irMPK4) – with low water-use efficiency (WUE) – to study the scale at which water-use traits alter intraspecific population yields. In the field and glasshouse, we observed overyielding in populations with low percentages of irMPK4 plants, unrelated to water-use phenotypes. Paired-plant experiments excluded the occurrence of overyielding effects at the neighbor scale. Experimentally altering field arbuscular mycorrhizal fungal associations by silencing the Sym-pathway gene *NaCCaMK* did not affect reproductive overyielding, implicating an effect independent of belowground AMF interactions. Additionally, micro-grafting experiments revealed dependence on shoot-expressed *MPK4* for *N. attenuata* to vary its yield per neighbor presence. We find that variation in a single gene, *MPK4*, is responsible for population overyielding through a mechanism, independent of irMPK4's WUE phenotype, at the aboveground, population scale.

**\*For correspondence:**
baldwin@ice.mpg.de

**Present address:** [†]Plant Pathogen Interaction, National Institute of Plant Genome Research, Aruna Asaf Ali Marg, New Delhi, Delhi; [‡]CONACYT-Consorcio de Investigación Innovación y Desarrollo para las Zonas Áridas (CIIDZA), Instituto Potosino de Investigación Científica y Tecnológica A. C., San Luis Potosí, México; [§]Geography, University of Zurich, Zürich, Switzerland

## Introduction

Plant trait diversity is known to increase the productivity and stability of plant populations (*Cardinale et al., 2012*; *Isbell et al., 2017*; *Isbell et al., 2015*; *Liang et al., 2016*; *Loreau and de Mazancourt, 2013*). These effects may be produced from single gene variation among individuals: previously, wheat populations comprising of two lines varied only in their resistance to powdery mildew through single-gene modifications outperformed monoculture populations, even when the monocultures were comprised of resistant individuals (*Zeller et al., 2012*). Additionally, forward genetics research recently identified a genetic locus associated with this diversity-productivity relationship (*Wuest and Niklaus, 2018*). However, establishing the mechanism that confers population productivity from a single genetic basis is difficult due to the complexity of plant population experiments. When individuals varying in traits or in loci of interest are planted in pairs, it is possible to conclude that their responses are due to the neighbor (*Gibson et al., 1999*; *Harper, 1977*). When planted in populations, however, it is unclear whether a plant responds only to properties of its direct neighbors or of the entire population (*Gibson et al., 1999*; *Radosevich, 1987*). The scale, or the hierarchical level in biological organization (*Allen and Starr, 1982*), at which diversity-productivity effects are constrained has not yet been determined, despite their importance in the study of complex population interactions (*Schneider, 2001*). Here, we define and investigate two spatial scales within populations that can be responsible for changes in total yield (*Figure 1*): the neighbor scale, where responding individuals (RIs) are constrained to change their growth and yield (quantified in biomass and fitness correlates, that is flowers and seed capsules) only in response to direct neighbors that differ in trait expression (divergent plants); and the population scale, where RIs can respond to the total composition of divergent plants in the entire population, creating a change in

**eLife digest** Whether on farmland or in a forest, plants do not grow in isolation. Plants compete with their neighbors over limited space and resources, and individual plants respond to this competition in different ways by changing how much they grow and how they use resources. The efficiency with which crop plants use water, for example, is one trait that is dramatically influenced by neighboring plants and is of increasing concern given the warming climate.

Understanding the effects of interactions between individual plants in a population as a whole is complicated, especially in natural plant communities where neighbors are often from different species. For this reason, McGale et al. took a different approach and looked at neighbors that were all from the same species and differed only in the activity of a single gene. The species in question was coyote tobacco, a plant that is native to western North America.

McGale et al. used genetic engineering to silence a gene called *MPK4*, which was known from previous studies to have the effect of reducing water-use efficiency. Some of these 'water-inefficient' plants were then grown in mixed populations with plants that had normal levels of *MPK4*. In experiments conducted both in a glasshouse and at a field station in the Utah desert, McGale et al. found that populations with a low percentage of the *MPK4*-silenced plants were actually more productive than 'monocultures' that were all one type or the other.

Further analysis showed that the increase in productivity did not depend on the different soil nutrient or water use of the different populations, or even the density of the plants in the populations. Pairs of plants grown in single pots essentially ruled out any interactions between immediate neighbors being responsible for the increased productivity, suggesting that that effect must instead emerge at the level of the population.

Perhaps unexpectedly, McGale et al. also found that the *MPK4*-silenced plants and control plants did not actually differ in how they used water when grown in the field (previous studies had all been conducted in glasshouses), indicating that this trait also could not explain the observed population-level effect. Finally, experiments that involved grafting the shoots of one plant onto the roots of another suggested that the effect most likely comes from the aboveground parts of the plant.

Ecologists have previously noted that more diverse populations typically have higher productivity. This new finding that a small percentage of slightly different plants in an otherwise uniform population can increase overall productivity will likely to be of special interest to researchers looking to boost the efficiency of agricultural ecosystems. Also, since *MPK4* is highly conserved, and thus likely to be found in many plant species, this could be an interesting trait with which to study the interactions of natural plant communities.

total population yield in direct proportion to RI abundance (*Crawford and Rudgers, 2012*; *Hughes et al., 2008*; *Smith and Knapp, 2003*).

Water-use traits are known to result in changes in total population yield (*Caldeira et al., 2001*; *Comas et al., 2013*; *Forrester, 2015*; *Kimball et al., 2014*; *Marguerit et al., 2014*; *Wang et al., 2016*; *Wu et al., 2016*). However, the scale at which this occurs (neighbor or population) remains unknown. WUE naturally varies among individuals, both within and among species (*Anderegg, 2015*; *Donovan et al., 2007*; *Heschel et al., 2002*; *Tortosa et al., 2016*; *Yoo et al., 2009*) and intraspecific variation in WUE traits can be as great as interspecific variation (*Messier et al., 2010*). Yield effects resulting from intraspecific WUE trait variation are of considerable agricultural interest (*Dutra et al., 2018*; *Sreeman et al., 2018*). Interestingly, WUE traits of some trees species alter the photosynthetic parameters and survival of neighboring trees (*Bunce et al., 1977*), suggesting potential neighbor-scale responses that can dramatically influence the yield of populations. However, studies have not pursued how WUE trait variation cause either population-scale or neighbor-scale responses that are responsible for changes in population growth and yield due to the complications that emerge in studying variations in WUE phenotypes.

To adequately study the scale at which RIs respond to variation in WUE of neighbors, one needs to anticipate several factors that would confound the analysis. WUE is calculated as the ratio of carbon assimilation to transpirational water loss, and WUE phenotypes typically result from altered stomatal function that increases plant transpiration. As the frequency of plants with low WUE (high

transpiration) increases in a population, the availability of soil water to the population is known to decrease proportionally (*Zea-Cabrera et al., 2006*). RIs may change their growth and yield in response to differences in soil water availability, rather than to the abundance of low-WUE plants in a population. Therefore, controlling for soil water availability independently of the frequency of plants with different WUE traits is essential for the analysis. The ecological relevance of variation in WUE traits is best evaluated in field populations, but standardizing water availability across populations in the field is rarely possible and thus combining inferences from field and glasshouse experiments, where soil water availability can be controlled using gravimetrically controlled watering, provides a useful way forward. Plants that vary in WUE as a result of single-gene manipulations greatly facilitate investigations into the scales at which yield responses are realized in populations. Here, we use isogenic plants, silenced in the expression of a single gene that profoundly influences stomatal behavior, to explore the scale at which WUE variation influences population yields.

Mitogen-activated protein kinases (MAPKs) are part of a conserved signaling cascade essential in eukaryotes. The downstream targets of this phosphorylation cascade, such as transcription factors, enable specific plant responses through changes in plant growth and development (*Xu and Zhang, 2015*). Mitogen-activated protein kinase 4 (MPK4) in *Nicotiana attenuata* and its homologues, MPK12 in *Arabidopsis thaliana* and MPK4/MPK4L in *N. tabacum*, have been implicated in responses to herbivore damage (*Gomi et al., 2005*; *Hettenhausen et al., 2013*; *Yanagawa et al., 2016*), bacterial inoculation (*Hettenhausen et al., 2012*), changes in exogenous and endogenous abscisic acid (ABA) and hydrogen peroxide levels (*Des Marais et al., 2014*; *Hettenhausen et al., 2012*;

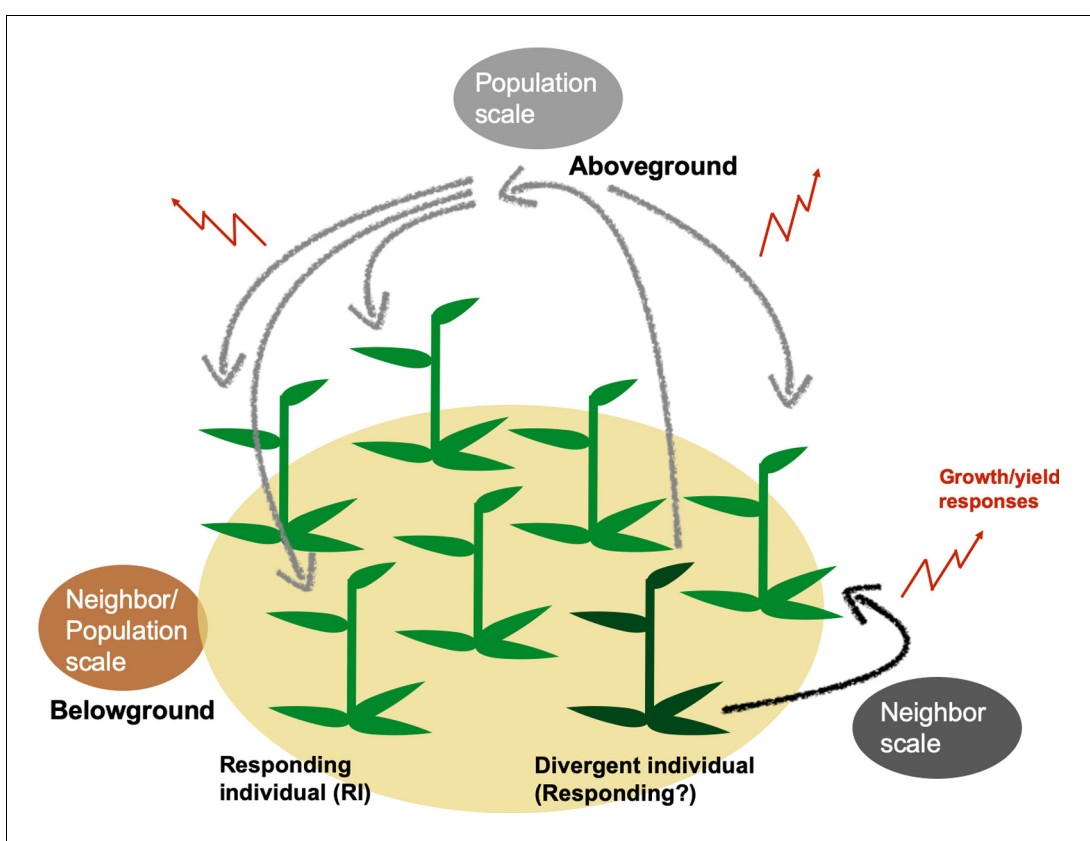

**Figure 1.** Genetically divergent plants can alter population yields through plant neighbor responses generated at neighbor or population scales. Genetic variation within a population can change population growth and yield by altering individuals' outputs either in localized areas within populations (neighbor scale) or across all plants of a population (population scale). At the neighbor scale (black), a divergent individual (dark green) may elicit responses only in immediate neighbors (responding individuals, RIs). RIs' responses at either spatial scale may include changes in growth and yield, which can cumulatively change a population's growth and yield (red). Responses to divergent individuals could be caused by above- (black, gray) and/or belowground interactions (brown) among plants in population.

*Jammes et al., 2009*), vapor pressure deficits (*Des Marais et al., 2014*) and ozone levels (*Gomi et al., 2005*; *Yanagawa et al., 2016*). Most of these responses involve the regulation of stomatal structure and function: silencing *NaMPK4* or *NtMPK4/L* by RNA interference (Na-irMPK4 and Nt-MPK4/L-IR, respectively) or knocking out *AtMPK12* (At-*mpk12*) results in plants with larger stomata and stomatal apertures, and varying disruptions in stomatal closure (*Des Marais et al., 2014*; *Gomi et al., 2005*; *Hettenhausen et al., 2012*; *Marten et al., 2008*; *Yanagawa et al., 2016*).

The alteration of stomatal phenotypes by *MPK4/12* expression strongly influences WUE. Na-irMPK4, Nt-MPK4/L-IR and At-*mpk12* all have increased transpiration rates which can be attributed to increased stomatal conductance (*Des Marais et al., 2014*; *Gomi et al., 2005*; *Hettenhausen et al., 2012*; *Yanagawa et al., 2016*). For Na-irMPK4 and At-*mpk12*, this increase in transpiration rates has been shown to dwarf the associated increases in assimilation rates, resulting in low WUE (*Des Marais et al., 2014*; *Hettenhausen et al., 2012*). However, previous glasshouse studies that tested whether the presence of *MPK4/12*-derived WUE phenotypes results in individual growth and yield effects in paired-plant-in-a-pot interactions did not control for soil water availability (*Des Marais et al., 2014*; *Hettenhausen et al., 2012*). To our knowledge, no study has investigated whether variation in the abundance of a low WUE trait, generated from the silencing of a single gene, affects population yield; similarly unstudied is the scale at which this might occur. Here, we conduct such a study using experimental *N. attenuata* populations in both the glasshouse and field.

The wild tobacco *N. attenuata* grows in xeric habitats in the western United States, where water limitation and WUE are selective factors throughout the growing season. *N. attenuata* typically grows in genetically diverse populations, in near-monocultures (*Baldwin and Morse, 1994*; *Baldwin et al., 1994*), and is known to respond differently when paired with genetically varied intraspecific neighbors: a *N. attenuata* accession collected in Utah (UT) sharing a pot with an accession collected in Arizona (AZ) produces significantly smaller stalks than when sharing a pot with another UT plant (*Glawe et al., 2003*). *N. attenuata* plants also naturally interact with AMF in the field, establishing networks of connected plants that have the potential to significantly change individuals' responses to neighbors or populations (reviewed in *Oelmüller, 2019*). AMF are known to change soil water availability and transport among individuals in populations based on each individual's ability to interact with AMF (*Egerton-Warburton et al., 2007*; *Reynolds et al., 2003*; *Yang et al., 2013*). Additionally, AMF have been shown to significantly affect plant-plant interactions in populations, as they can transfer nutrients, defense signals and allelopathic chemicals (*Ferlian et al., 2018*; *Gorzelak et al., 2015*; *Song et al., 2019*). Manipulating the ability of a population to interact through an AMF network can provide a means of dramatically altering belowground interactions and narrow the potential causes of population yield changes due to within- (or between-) species plant diversity (*Figure 1*). Calcium and calmodulin-dependent kinase (CCaMK) is required for successful plant symbiosis with AMF (*Lévy et al., 2004*), and the abrogation of *CCaMK* expression provides a valuable tool to disconnect plants from AMF networks in the field (*Groten et al., 2015*). Field plantations of transgenic *N. attenuata* crossed with *CCaMK*-deficient transgenic lines (irCCaMK) in the plant's native habitat, the Great Basin Desert, allow for the study of population growth and yield effects resulting from trait variation, as well as the scales and tissues in which this variation occurs. Here, we use a single gene manipulation to create variation in WUE in the background of irCCaMK lines to separate the effects of AMF-mediated interactions in our analysis of the spatial scales at which variation in WUE traits influence population yields.

We investigated the spatial scales at which variation in abundance of low WUE *N. attenuata* plants, generated by the abrogation of *MPK4* expression, change total population growth and yield. We used a previously characterized irMPK4 line (*Hettenhausen et al., 2012*) and varied the percentages of this line in field populations with empty-vector (EV) control plants, both crossed with either irCCaMK and EV lines, to manipulate connectivity to the AMF network. We observed increased yields, referred to as overyielding, in populations with low percentages of *MPK4*-deficient plants ('low-irMPK4'), due primarily to increases in EV plant yield. To exclude soil water availability effects, we grew homozygous irMPK4 and EV lines in glasshouse populations under equal water availability and again observed overyielding in low-irMPK4 populations due to increases in EV yield. We further tested responses at the neighbor scale by growing mono- and mixed-genotype pairs of EV and irMPK4 under conditions of equal water availability and found no changes in growth or yield between pair types. We analyzed the yield of individuals with different configurations of immediate neighbor genotypes in our glasshouse populations, but these also did not explain changes in

individual yields. From these results, we conclude that neighbor-scale responses are unlikely to be responsible for the overyielding phenomena. In the glasshouse, changes in EV plants' photosynthetic parameters did not explain the yield increases in low-irMPK4 populations and importantly, EV and irMPK4 plants did not differ in their WUE phenotypes in field populations. Therefore we inferred that irMPK4's WUE phenotype was not responsible for the observed overyielding at the population scale. In neighbor absence-presence paired tests, we observed that EV plants change their growth and yield when planted with or without a neighbor, while interestingly, irMPK4 plants do not. In similar experiments with EV shoots micro-grafted to irMPK4 roots, *MPK4* expression in the shoot could remediate this effect, demonstrating a shoot-localized role for *MPK4* in *N. attenuata's* ability to alter

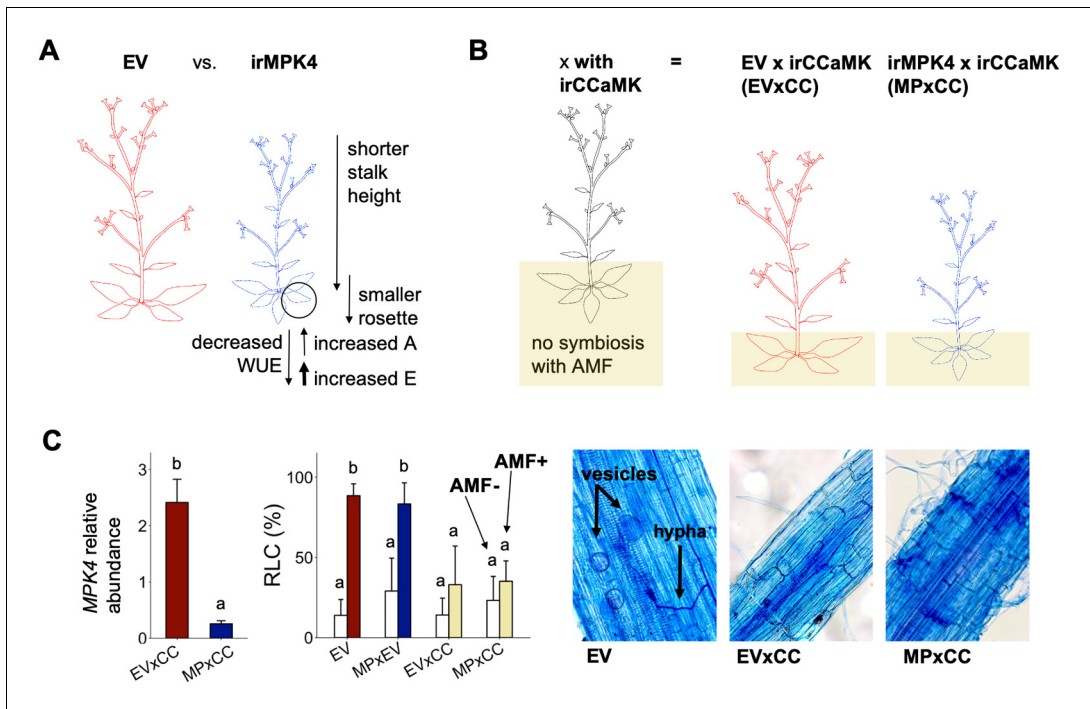

**Figure 2.** Characterization of EV, irMPK4, irMPK4xEV, EV x irCCaMK and irMPK4 x irCCaMK plants. (**A**) A schematic summary of the findings of *Hettenhausen et al. (2012)*: irMPK4 *Nicotiana attenuata* plants, silenced in *mitogen-activated protein kinase 4*, have disrupted stomatal control resulting in increased rates of leaf transpiration (**E**) which surpass the smaller increases in rates of leaf carbon assimilation (**A**) and therefore decrease water-use efficiency (WUE) in comparison to empty-vector (EV) plants. irMPK4 stalks and rosettes are smaller than those of EV. (**B**) A schematic demonstrating a method established by *Groten et al. (2015)* to control arbuscular mycorrhizal association in the field: irCCaMK *N. attenuata* plants, silenced in *calcium and calmodulin-dependent protein kinase*, are crossed with EV and irMPK4 to create EV x irCCaMK (EVxCC) and irMPK4 x irCCaMK (MPxCC) lines hemizygous for each of the transgenes and are not able to associate with arbuscular mycorrhizal fungi (AMF). (**C**) *Mitogen-activated protein kinase 4* (*MPK4*) transcript abundances, calculated relative to a housekeeping gene, in hemizygous MPxCC and EVxCC plants (left panel, mean + CI, n = 9 for EV, 13 for irMPK4) compare with those of homozygous irMPK4 and EV plants (*Figure 2—figure supplement 1*). EVxCC and MPxCC roots inoculated with an arbuscular mycorrhizal fungus, *Rhizophagus irregularis* (AMF+), did not show significant increases in comparison to un-inoculated counterparts (AMF-) in root length colonization (RLC; center panel, mean + CI, n = 7 for EVxCC, n = 8 for MPxCC), in contrast to the strong colonization of EV plants (n = 8) and control hemizygous irMPK4 crosses: irMPK4xEV (MPxEV, n = 7–8). Vesicles and hyphae are visible in trypan blue-stained AMF+ EV roots, but not in AMF+ EVxCC and MPxCC roots.

The online version of this article includes the following source data and figure supplement(s) for figure 2:

**Source data 1.** Source data for *Figure 2*.

**Figure supplement 1.** *MPK4* transcript abundance relative to a housekeeping gene (+ 95% CI) and silencing efficiency in homozygous EV and irMPK4 shoots (top) and roots (bottom) which were grafted together for the glasshouse grafting pair experiment (n = 8 for EV/EV, 9 for EV/irMPK4, 6 for irMPK4/irMPK4).

**Figure supplement 1—source data 1.** Source data for *Figure 2—figure supplement 1*.

yield in the presence of a neighbor. Additionally, manipulating field population's AMF connectivity did not change observed reproductive overyielding, denoting a lack of belowground influence on this effect. From these results, we suggest a novel function of shoot *MPK4* in mediating *N. attenuata*'s yield response to neighbors, unrelated to WUE, which in low-irMPK4 populations may result in reproductive overyielding.

## Results

### irMPK4 x irCCaMK crosses are silenced in *MPK4* and abrogated in AMF associations

*N. attenuata* plants silenced in the expression of *MPK4* (irMPK4) have a low water-use efficiency (WUE) phenotype in comparison to empty-vector (EV) control plants in the glasshouse (*Figure 2A*). The loss of stomatal control increases transpiration rates to levels that surpass the increases in assimilation rates, consequently decreasing WUE, calculated as the ratio of assimilation:transpiration rates (*Hettenhausen et al., 2012*).

Populations of plants growing in the field are commonly interconnected by arbuscular mycorrhizal fungal (AMF) networks that are known to influence access to water and nutrients in the plant rhizosphere (*Egerton-Warburton et al., 2007*; *Reynolds et al., 2003*; *Yang et al., 2013*), as well as within-population plant neighbor responses (*Ferlian et al., 2018*; *Gorzelak et al., 2015*; *Song et al., 2019*). As silencing the expression of *NaCCaMK* disconnects plants from AMF networks (*Groten et al., 2015*), we crossed isogenetic, homozygous irCCaMK plants with homozygous EV and irMPK4 lines to generate hemizygous EV x irCCaMK (EVxCC) and irMPK4 x irCCaMK (MPxCC) lines (*Figure 2B*), which were used for field experiments. The hemizygous crosses retained the levels of *MPK4* silencing of the homozygous irMPK4 lines: MPxCC showed an 87% reduction of *MPK4* transcript accumulation relative to EVxCC in the field (*Figure 2C*), whereas irMPK4 had 83% silencing efficiency relative to EV in the glasshouse (*Figure 2—figure supplement 1*). To evaluate the abrogation of AMF associations under controlled conditions, we grew the EVxCC and MPxCC crosses in the glasshouse with and without live AMF inoculum (*Rhizophagus irregularis*) and compared their AMF colonization characteristics to that of EV and a hemizygous irMPK4xEV (MPxEV) control cross. While EV and MPxEV were highly colonized in comparison to non-inoculated controls (*Figure 2C*, LM, *emmeans*$_{(EV, AMF-(n = 8) to AMF+(n = 8))}$, t = $-8.894$, p = <0.0001; *emmeans*$_{(MPxEV, AMF-(n = 7) to AMF+(n = 8))}$, t = $-6.253$, p = <0.0001), both EVxCC (*emmeans*$_{(EVxCC, AMF-(n = 7) to AMF+(n = 7))}$, t = $-2.105$, p=0.4251) and MPxCC (*emmeans*$_{(MPxCC, AMF-(n = 8) to AMF+(n = 8))}$, t = $-1.417$, p=0.8453) did not differ from un-inoculated controls in root length colonization (RLC). Trypan blue-staining of roots showed the establishment of vesicles and hyphae in EV, but not in EVxCC and MPxCC plants (*Figure 2C*). From these results, we conclude that the hemizygous crosses retain their *MPK4* silencing and do not associate with AMF.

### Populations with low percentages of *MPK4*-deficient plants show overyielding in both the field and glasshouse

In order to evaluate if the percentage of *MPK4*-deficient plants influences population yield under field conditions, growth and yield of EVxCC and MPxCC individuals in populations with varying percentages of MPxCCs (0, 25, 75, 100%; *Figure 3A*; *Figure 3—figure supplement 1*) were measured and analyzed using de Wit replacement diagrams (*Figure 3B–G*; *Figure 3—figure supplement 2*; *de Wit, 1960*; *Harper, 1977*). Increases in yield, referred to as overyielding, were observed in the relative yield totals (RYTs) of 25% MPxCC populations in stalk height (*Figure 3C*), shoot and root biomass (*Figure 3D–E*), and unripe and ripe seed capsule values (*Figure 3F–G*). This overyielding was due only to increases in EVxCC plants: in 25% irMPK4 populations, cumulative EVxCC plant trait values exceeded their predicted values based on their performance in monoculture. MPxCC plant trait values did not differ from their monoculture values in any population type. However, the increase in the cumulative EVxCC trait values in the replacement diagram was not reflected in significant differences between means of EVxCC individuals in 25% MPxCC populations versus in other population types (*Figure 3—figure supplement 2*), emphasizing the role of incremental benefits observable at the population-scale rather than in the performance characteristics of each individual in the population.

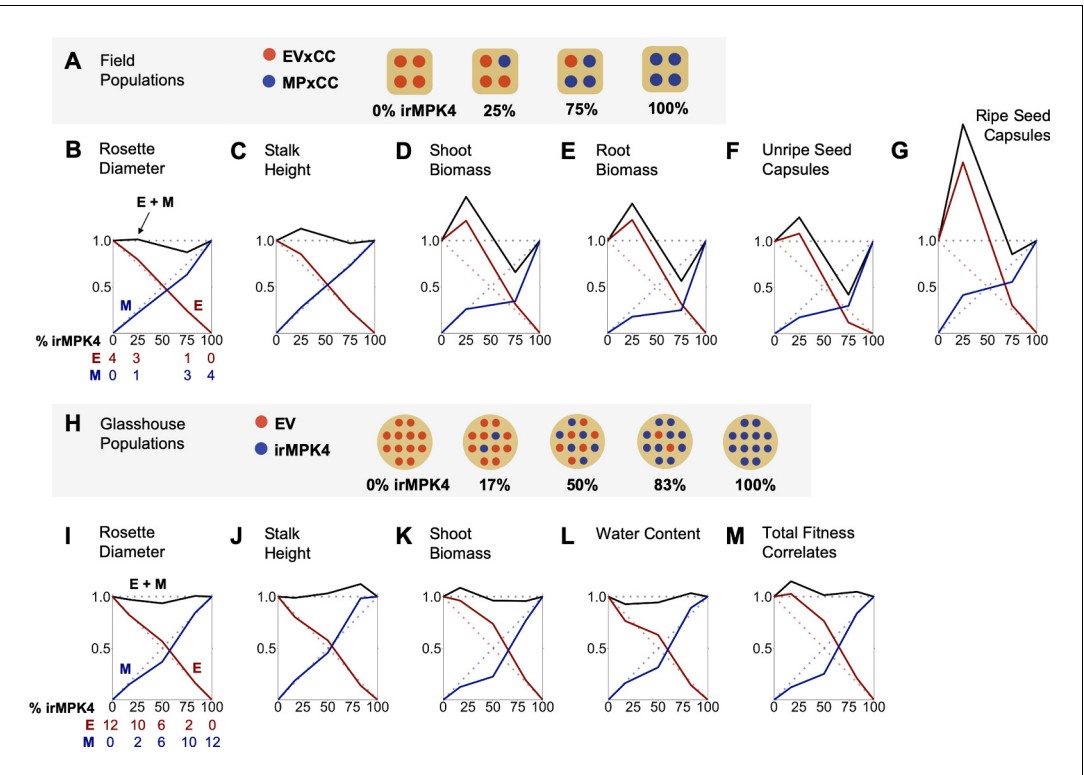

**Figure 3.** In the field and glasshouse, populations with low percentages of *MPK4*-deficient plants show overyielding. (**A**) Field populations of four plants around a central water dripper were planted with varying percentages of EV and irMPK4 (MP) plants crossed with irCCaMK (CC) to abolish interaction with arbuscular mycorrhizal networks: four EVxCC (0%, n = 12), three EVxCC and one MPxCC (25%, n = 20), one EVxCC and three MPxCC (75%, n = 20), or four MPxCC (100%, n = 20) plants (for additional details of the experimental set-up see *Figure 3—figure supplement 1*). (**B – G**) Replacement diagrams show relative (**B**) rosette diameters (n = 11–35); (**C**) stalk heights (n = 6–29); (**D**) shoot biomasses (n = 10–31); (**E**) root biomasses (n = 8–31); (**F**) unripe seed capsules (n = 4–14); and (**G**) ripe seed capsules (n = 3–17) of EVxCC (E, red) and MPxCC (M, blue) plants in 0–100% irMPK4 field populations. Relative growth and yield for each genotype was calculated as: (trait mean in mixture * # of plants in mixture)/(trait mean in monoculture * 4). Means and error structures are shown in *Figure 3—figure supplement 2*. Relative yield totals of the populations (RYT, black) are calculated as E + M. Dotted lines indicate predicted yields from plants in monocultures. (**H**) Glasshouse populations of 12 plants were planted with varying percentages of irMPK4 plants: 12 EV (0%), 10 EV and 2 irMPK4 (17%), 6 EV and 6 irMPK4 (50%), 2 EV and 10 irMPK4 (83%), or 12 irMPK4 (100%). Each population was watered in proportion to its daily water consumption to ensure equal water availability across all populations (for additional details of the experimental set-up see *Figure 3—figure supplement 5B* and *Water treatments* in Materials and methods). (**I – M**) Replacement diagrams show relative (**I**) rosette diameters (n = 11–35); (**J**) stalk heights (n = 21–41); (**K**) shoot biomasses (n = 21–41); (**L**) water contents (n = 22–41); and (**M**) total reproductive yield measured as counts of fitness correlates (n = 19–44) from EV (E, red) and irMPK4 (M, blue) plants in 0–100% irMPK4 glasshouse populations. Relative growth and yield for each genotype was calculated as: (trait mean in mixture * # of plants in mixture)/(trait mean in monoculture * 12). Means and error structures are shown in *Figure 3—figure supplement 6*. RYTs (black) are calculated as E + M. Dotted lines indicate predicted yields from monocultures.

The online version of this article includes the following source data and figure supplement(s) for figure 3:

**Source data 1.** Source Data for *Figure 3*.

**Figure supplement 1.** Layout of field population experiment at the Lytle Ranch Reserve field plot ('Snow Plot', Utah) from an (**A**) aerial view (pink, Google Images), from a (**B**) side view (pink, picture by E. M.), and as a (**C**) schematic of all planted populations, with an example population at harvest, 53 days post planting (dpp; inset, picture by EM).

**Figure supplement 2.** Growth and yield of EV and irMPK4 individuals crossed with irCCaMK (EVxCC and MPxCC, respectively) compared when planted in the varying field population types (see *Figure 3A*): A) rosette diameter (mean + CI, n = 11–35; 13 dpp), (B) stalk height (mean + CI, n = 6–29; 23 dpp), (C) shoot biomass (mean + SE, *Figure 3 continued on next page*

*Figure 3 continued*

n = 10–31; 46–53 dpp), (D) root biomass (mean + SE, n = 8–31; 46–53 dpp), (E) unripe seed capsules (mean + SE, n = 4–14; 46–53 dpp) and F) ripe seed capsules (mean + SE, n = 3–17; 46–53 dpp).

**Figure supplement 2—source data 1.** Source Data for *Figure 3—figure supplement 2*.

**Figure supplement 3.** Soil element concentration (mean + CI, y-axis, n = 3) of (A) total carbon, (B) total nitrogen, (C) inorganic carbon, (D) organic carbon, (E) copper, (F) iron, (G) phosphorus, (H) potassium, and (I) zinc, taken from soil cores 5, 15 and 30 cm below the center dripper of each population type (x-axis).

**Figure supplement 3—source data 1.** Source Data for *Figure 3—figure supplement 3*.

**Figure supplement 4.** Soil moisture (%) of soil cores taken 5, 15 and 30 cm below the center dripper of each population type over 9 days, either (A) cumulatively across the plot or (B) divided into the 'Dry' and 'Wet' subsections (see *Water treatments* in Materials and methods), with corresponding regression analysis results.

**Figure supplement 4—source data 1.** Source Data for *Figure 3—figure supplement 4*.

**Figure supplement 5.** Layout of glasshouse population experiment as pictured from (A) above the glasshouse table (picture by DM), and above a single example pot (inset, picture by EM).

**Figure supplement 6.** Growth and yield of EV and irMPK4 individuals compared when planted among the varying glasshouse population types (see *Figure 3H*): (A) rosette diameter (mean + CI, n = 11–35; 30 dpp), (B) stalk height (mean + CI, n = 21–41; 30 dpp), (C) shoot biomass (mean + CI, n = 21–41; 50 dpp), (D) water content (mean + CI, n = 22–41; 50 dpp), (E) total reproductive yield measured as counts of fitness correlates (buds, flowers, unripe and ripe seed capsules; mean + CI, n = 19–44; 50 dpp).

**Figure supplement 6—source data 1.** Source Data for *Figure 3—figure supplement 6*.

**Figure supplement 7.** Characterization experiment comparing EV to EV x irCCaMK (EVxCC) and irMPK4xEV (MPxEV) to irMPK4 x irCCaMK (MPxCC) in (A) water loss per day (g ± CI, n = 15–16), (B) fresh shoot biomass (g + CI, n = 5–7), and (C) fresh root biomass (g + CI, n = 6–7).

**Figure supplement 7—source data 1.** Source data for *Figure 3—figure supplement 7*.

**Figure supplement 8.** Yara ZIM-probe leaf turgor measurements (mean kPa, representative recordings of three replicate measurements are shown).

**Figure supplement 8—source data 1.** Source Data for *Figure 3—figure supplement 8*.

---

Plants with low WUE are thought to increase the flow of water-soluble nutrients to the immediate area around their roots as a consequence of excessive transpiration rates (*del Amor and Marcelis, 2005*; *Zea-Cabrera et al., 2006*). Therefore, we collected soil cores at 5, 15 and 30 cm below the center of each population type and quantified total carbon ($C_{total}$), nitrogen (N), inorganic carbon ($C_{inorg}$), organic carbon ($C_{org}$), copper (Cu), iron (Fe), potassium (K), phosphorus (P) and zinc (Zn) concentration (*Figure 3—figure supplement 3*). At each depth, there were no significant differences among populations for any nutrient except for $C_{inorg}$ which was slightly increased at 5 cm depth in the 0% populations (EVxCC monoculture), at 15 cm in 75% MPxCC populations, and at 30 cm in 100% MPxCC populations (*Figure 3—figure supplement 3C*). We observed no increases in any inorganic nutrient at any soil depth in the 25% MPxCC populations (*Figure 3—figure supplement 3*). Furthermore, the percentage of MPxCC plants in populations did not significantly predict soil moisture at any sampling depth (*Figure 3—figure supplement 4A*). From these results, we conclude that increasing the percentage of MPxCC plants in populations under field conditions leads to a non-additive trend in population yield, unrelated to soil moisture and inorganic nutrient availability, with overyielding occurring in 25% irMPK4 populations.

To further evaluate whether the water-use phenotype of irMPK4 plants contributed to differences in water and nutrient availability for populations in ways that were undetectable in the field, we created populations in the glasshouse with increasing percentages of irMPK4 (0, 17, 50, 83% and 100%; *Figure 3H*; *Figure 3—figure supplement 5A*) in which we experimentally controlled for water availability among populations (*Figure 3—figure supplement 5B*; see *Water treatments* in Materials and methods). Replacement diagrams were again used to analyze cumulative growth and yield of EV and irMPK4 plants in varying population types (*Figure 3I–M*; individual means in *Figure 3—figure supplement 6*). The analysis revealed overyielding in shoot biomass and total fitness correlates (reproductive yield) of low-irMPK4 populations (17%; *Figure 3K,M*), consistent with the field results. Due to the controlled watering schema of the glasshouse experiment, we conclude that this overyielding effect is independent of population water availability.

## Overyielding in low-irMPK4 populations does not occur at the neighbor scale

To test if overyielding in low-irMPK4 field and glasshouse populations (*Figure 3*) resulted from neighbor interactions of EV and irMPK4 plants, we investigated the growth and yield of EV and irMPK4 in monoculture and mixed pairs (*Figure 4A*), again under conditions of equal water availability (*Figure 4B*). Replacement diagrams revealed no evidence of overyielding in any of the measured growth and yield parameters for the mixed pairs (*Figure 4G–J*). We conclude that for EV plants having one irMPK4 neighbor was not sufficient to produce the overyielding response we observed in low-irMPK4 populations.

Varying local configurations of irMPK4 neighbors could also cause neighbor-scale overyielding in EV plants, a property we would not observe in our paired plant experiment. Therefore, in the glasshouse population experiment (*Figure 3H*), we analyzed growth and fitness measurements of centrally located EV individuals with four direct neighbors. In 0% irMPK4 populations, all four neighbors were EV plants, in 17% irMPK4 populations, two were EV and two were irMPK4 plants, and in 50% and 83% irMPK4 populations, all four were irMPK4 plants. We observed that only in 50% irMPK4 populations, EV plants with four irMPK4 neighbors produced significantly higher growth and yield in comparison to EV plants grown in 0% irMPK4 populations (*Figure 3—figure supplement 6*; *Supplementary file 1*). However, 50% irMPK4 populations did not show overyielding (*Figure 3I–M*), likely because irMPK4 plants simultaneously had significantly smaller rosettes, water contents, and yields compared with 100% irMPK4 monocultures (*Figure 3—figure supplement 6*; *Supplementary file 1*). Importantly, EV plants grown in 50% and 83% irMPK4 populations, with the same irMPK4 direct neighbor configuration, did not show consistent changes in growth and yield compared to monocultures. These results are consistent with the inference that overyielding does not occur at the neighbor scale.

## *MPK4* is necessary for *N. attenuata* to change their growth and yield when planted with a neighbor

While an immediate neighbor response is not mediating the population overyielding response, we observed strong changes mainly in EV plants across our different population experiments. To test whether EV and irMPK4 plants respond differently to the presence of a neighbor, we included EV and irMPK4 planted as singles in our paired-pot experiment (*Figure 4A*). We compared the growth and yield to single plants with individuals in mono- and mixed-culture pairs. Water contents of EV and irMPK4 plants did not differ, whether planted alone or in pairs (*Figure 4E*; *Table 1*), indicating equal water availability in the two potting types. EV plants with an EV or irMPK4 neighbor had smaller rosettes, shoot biomass, and reproductive yield than when planted alone (*Figure 4C–D and F*; *Table 1*). However, this reduction was independent of the neighbor's genotype. In contrast, irMPK4 plants showed no differences in their rosette growth, shoot biomass, or yield when planted in pairs as compared to being grown alone (*Figure 4C–F*). From these results, we conclude that *MPK4* is required for *N. attenuata*'s growth and yield responses to a neighbor.

## EV and irMPK4 photosynthetic phenotypes do not explain overyielding in the glasshouse

To determine if the WUE phenotypes of EV and irMPK4 plants in glasshouse and field populations change with the percentage of *MPK4*-deficient plants, potentially causing overyielding in low-irMPK4 populations (*Figure 3*), we measured leaf photosynthetic parameters (assimilation rate, transpiration rate, stomatal conductance) and calculated the WUE of all individuals in both glasshouse and field experiments.

In the glasshouse paired experiment, all measured leaf photosynthetic parameters of EV and irMPK4 plants in single pots were as previously reported (*Figure 2A*), with irMPK4 plants having significantly higher assimilation rates, transpiration rates, and stomatal conductance than EV plants, and significantly lower WUE (*Figure 5A*; *Table 2*). When planted in pairs, EV and irMPK4 plants' assimilation rates, transpiration rates, and stomatal conductance were not significantly different in monoculture (red and blue shadings, respectively) versus in mixed culture (purple shading). EV plants had significantly lower WUE in mixed versus monoculture (*Figure 5A*; *Table 2*), whereas irMPK4 plants showed no significant change in WUE across the planting types.

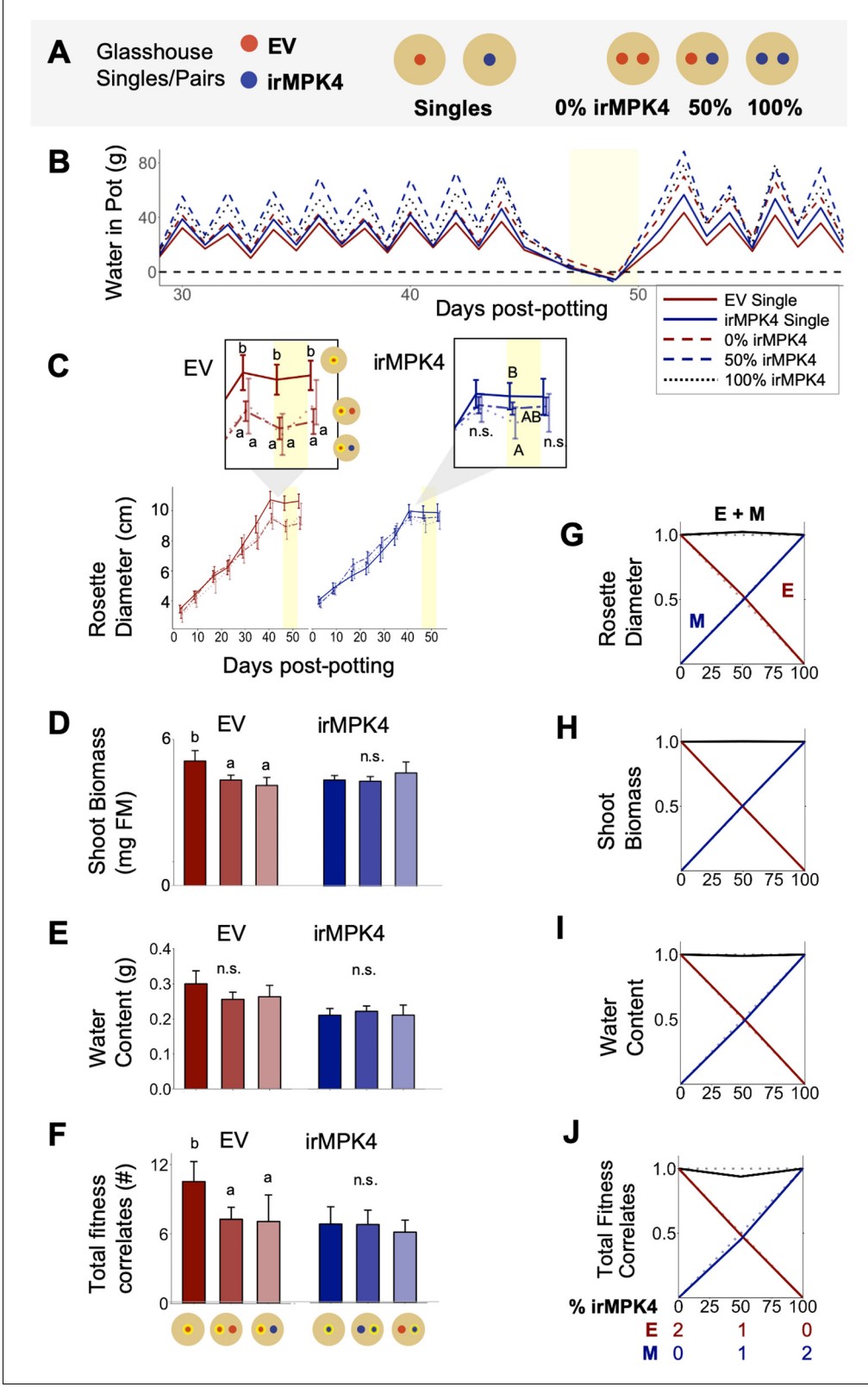

**Figure 4.** In the glasshouse under equal water availability, EV but not irMPK4 plants have reduced growth in the presence of a neighbor. (**A**) EV and irMPK4 were planted either as singles or in mono- or mixed-culture pairs. EV monoculture pairs have 0%, mixed-cultures have 50%, and irMPK4 monocultures have 100% irMPK4 plants. (**B**) All pots were watered based on daily individual consumption to ensure equal water availability (see *Water treatments*

*Figure 4 continued on next page*

*Figure 4 continued*

in Materials and methods): mean g of water per pot for each potting type (red, solid: EV Single; red, dashed: 0% irMPK4; blue, solid: irMPK4 Single; blue, dashed: 100% irMPK4; black, dotted: 50% irMPK4) immediately following a watering event (graphical peaks) and immediately preceding the next watering event (graphical troughs) are displayed. Withholding water for 2 days caused all pots to reach a state of no available water in the pot (yellow shading). (C – F) EV (red) and irMPK4 (blue) individual means in each pot type (Single: solid line; 0%/100%: dashed line; 50%: dotted line) for (C) rosette diameter (mean cm ±95% CI, n = 11–24; 3–53 days post potting, dpp), (D) shoot biomass (mean + CI, n = 10–22; 71dpp), (E) water content (mean + CI, n = 8–22; 71dpp), and (F) total reproductive yield measured as counts of fitness correlates (buds, flowers, unripe and ripe seed capsules; mean + CI, n = 9–22; 71dpp). Significant differences are presented within genotypes. Inset of (C): Significant differences in EV and irMPK4 rosette diameters among planting types are indicated for the last three time points of the main panel, within each genotype. To evaluate growth effects of the equal water availability (yellow shading), growth values before and after water was withheld are highlighted in the inset. (G – I) Replacement diagrams show (G) rosette diameters (n = 11–24; 53 dpp); (H) shoot biomasses (n = 10–22); (I) water contents (n = 8–22); and (J) total reproductive yield measured as counts of fitness correlates (n = 9–22) from EV (E, red) and irMPK4 (M, blue) plants in 0–100% irMPK4 glasshouse pairs, calculated as (trait mean in mixture*# of plants)/(trait mean in monoculture*2). Relative yield totals (RYT, black) are calculated as E + M. Means and error structures can be found in panels (C – F). Dotted lines indicate no deviations from yields in monocultures.

The online version of this article includes the following source data for figure 4:

**Source data 1.** Source data for *Figure 4*.

In the glasshouse population experiment, irMPK4 plants in 100% (blue shading) versus 50% (purple shading) irMPK4 populations showed no significant differences in any photosynthetic parameter (*Figure 5B*), which was consistent with the glasshouse paired experiment. EV plants in 0% (red shading) versus 50% (purple shading) irMPK4 populations were not significantly different from each other in any parameter except for significantly higher transpiration rates of EV plants in 50% irMPK4 populations compared with those in 0% irMPK4 populations (*Figure 5B*; LMER, EV: *emmeans* $0\%_{(n = 32)}$-$50\%_{(n = 16)}$, t = −3.744, p=0.0082). While the results of statistical comparisons of EV responses (0% versus 50% irMPK4) differ between the pair (*Figure 5A*) and population (*Figure 5B*) experiments, the effects on the means for the two experiments remained consistent: with EV transpiration rates increasing ($Pairs_{0\% \text{ to } 50\%}$: +0.49; $Populations_{0\% \text{ to } 50\%}$: +1.22) and WUE decreasing ($Pairs_{0\% \text{ to } 50\%}$: −3.56; $Populations_{0\% \text{ to } 50\%}$: −13.78).

In the glasshouse, EV plants in low-irMPK4 populations did not have significantly different photosynthetic parameter values compared with other population types (*Figure 5B*). In addition, photosynthetic parameters were measured at a pre-dawn (AM; *Figure 5—figure supplement 1*), which included dark-adapted chlorophyll fluorescence measurements (Fv/Fm) reflecting the maximum yield of the photosynthetic system (*Signarbieux and Feller, 2011*). The AM photosynthetic parameter values of EV plants in low-irMPK4 (17%) populations also did not significantly differ from EV plants in any other population type (*Figure 5—figure supplement 1*).

**Table 1.** *emmeans* contrasts of EV individuals in varying potting types for *Figure 4C–F*.*

| Model | Contrast | Trait | t-value | p-value |
|---|---|---|---|---|
| LM | EV Single$_{(n = 9)}$ − EV Mono$_{(n = 22)}$ | Water Content | 2.378 | 0.0614 |
| | EV Single$_{(n = 9)}$ −EV Mix$_{(n = 11)}$ | Water Content | 1.751 | 0.2047 |
| LM | EV Single$_{(n = 12)}$ − EV Mono$_{(n = 24)}$ | Rosette Diameter | 5.131 | <0.0001† |
| | EV Single$_{(n = 12)}$ − EV Mix$_{(n = 12)}$ | Rosette Diameter | 4.979 | <0.0001† |
| GLS | EV Single$_{(n = 11)}$ −EV Mono$_{(n = 22)}$ | Shoot Biomass | 4.196 | 0.0004† |
| | EV Single$_{(n = 11)}$ −EV Mix$_{(n = 10)}$ | Shoot Biomass | 4.531 | 0.0002† |
| LM | EV Single$_{(n = 11)}$ −EV Mono$_{(n = 21)}$ | Total Fitness Correlates | 3.848 | 0.0017‡ |
| | EV Single$_{(n = 11)}$ −EV Mix$_{(n = 11)}$ | Total Fitness Correlates | 3.323 | 0.0066‡ |

*extracted from linear (LM) or generalized least squares (GLS) models with significant ANOVA results.
†p value < 0.001; ‡p value < 0.01.

## In the field, EV and irMPK4 photosynthetic parameters are similar, regardless of AMF associations

In the field experiment, EVxCC and MPxCC plants that lacked the ability to associate with arbuscular mycorrhizal networks (AMF-), did not differ in any photosynthetic parameters, whether these were compared between genotypes or across population types (*Figure 5C*). To test if the ability to interact with an AMF network changes patterns of photosynthetic performance, we additionally analyzed photosynthetic parameters of EV and irMPK4xEV (MPxEV) plants that could interact with AMF networks (AMF+). Similar to the irCCaMK crosses (-AMF), EV and irMPK4 plants capable of associating

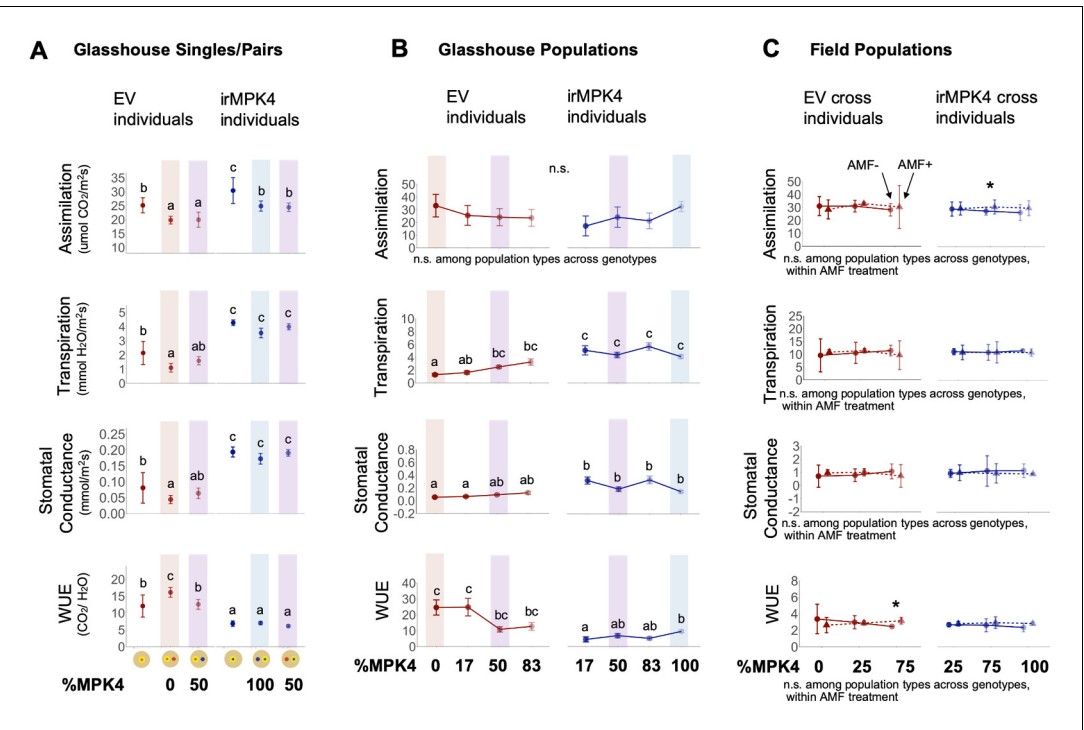

**Figure 5.** irMPK4 plants have low WUE in glasshouse, but not field experiments, regardless of AMF associations. (**A**) Assimilation rate, transpiration rate, stomatal conductance and water-use efficiency (WUE; mean ±CI, n = 3–8) of EV (red) and irMPK4 (blue) individuals from each planting type in the paired glasshouse experiment (see *Figure 4*) at 48 days post potting (dpp). To facilitate comparison of data to the population glasshouse experiment, EV in 0% irMPK4 populations (red shading), irMPK4 plants in 100% irMPK4 populations (blue shading) and both genotypes in 50% irMPK4 populations (purple shading) are highlighted. Significant differences are presented across genotypes. (**B**) Assimilation rate, transpiration rate, stomatal conductance and WUE (mean ±CI, n = 11–32) of EV (red) and irMPK4 (blue) individuals from each planting type in the population glasshouse experiment (*Figure 3H*) at 32 dpp. Measurements were taken between 12:00-14:00; additional pre-dawn measurements (4:00-6:00) are included in *Figure 5—figure supplement 1*. For comparison to the paired glasshouse experiment, EV and irMPK4 in 0% (red), 50% (purple) and 100% (blue) irMPK4 populations are highlighted. Significant differences are presented across genotypes. (**C**) Assimilation rate, transpiration rate, stomatal conductance and WUE (mean ±CI, n = 3) of EV (red, circle) and irMPK4xEV (blue, circle) individuals with the ability to associate with arbuscular mycorrhizal fungi (AMF, solid line), or EVxCC (red, triangle) and irMPK4xCC (blue, triangle) individuals without the ability to associate with AMF from the field population experiment. Measurements were performed at 34 dpp on irrigated plants ('Wet', see *Water treatments* in Materials and methods). Significant differences are presented both across genotypes, within AMF treatments (text below panels), or within the genotype and planting type, between AMF treatments (*: p<0.05).

The online version of this article includes the following source data and figure supplement(s) for figure 5:

**Source data 1.** Source Data for *Figure 5*.
**Figure supplement 1.** Pre-dawn photosynthetic measurements of EV (red) and irMPK4 (blue) plants in the various planting types of the population glasshouse experiment.
**Figure supplement 1—source data 1.** Source data for *Figure 5—figure supplement 1*.

**Table 2.** *emmeans* contrasts of EV to irMPK4 individuals planted as singles for **Figure 5A**\*.

| Model | Contrast | Trait | t-value | p-value |
|---|---|---|---|---|
| LME | EV Single$_{(n = 4)}$ – irMPK4 Single$_{(n = 3)}$ | Assimilation | −3.947 | 0.0134‡ |
| | EV Mono$_{(n = 7)}$ – EV Mix$_{(n = 4)}$ | Assimilation | −0.123 | 1.0000 |
| | rMPK4 Mono$_{(n = 8)}$ – irMPK4 Mix$_{(n = 4)}$ | Assimilation | 0.396 | 0.9985 |
| LME | EV Single$_{(n = 4)}$ – irMPK4 Single$_{(n = 4)}$ | Transpiration | −8.089 | <0.0001† |
| | EV Mono$_{(n = 8)}$ – EV Mix$_{(n = 4)}$ | Transpiration | −3.171 | 0.0527 |
| | irMPK4 Mono$_{(n = 8)}$ – irMPK4 Mix$_{(n = 4)}$ | Transpiration | −2.776 | 0.1104 |
| LME | EV Single$_{(n = 3)}$ – irMPK4 Single$_{(n = 4)}$ | SC | −8.089 | <0.0001† |
| | EV Mono$_{(n = 8)}$ – EV Mix$_{(n = 4)}$ | SC | −3.171 | 0.0527 |
| | irMPK4 Mono$_{(n = 8)}$ – irMPK4 Mix$_{(n = 4)}$ | SC | −2.776 | 0.1104 |
| LME | EV Single$_{(n = 4)}$ – irMPK4 Single$_{(n = 4)}$ | WUE | 6.394 | 0.0001† |
| | EV Mono$_{(n = 8)}$ – EV Mix$_{(n = 4)}$ | WUE | 3.723 | 0.0205‡ |
| | irMPK4 Mono$_{(n = 8)}$ – irMPK4 Mix$_{(n = 4)}$ | WUE | 3.203 | 0.0544 |

\*extracted from linear-mixed effect (LME) models with significant ANCOVA results SC = Stomatal Conductance; WUE = Water Use Efficiency.

†p value < 0.001; ‡p value < 0.05.

with AMF did not differ across population types and the two genotypes (**Figure 5C**). We further tested whether the AMF association could change photosynthetic parameters within a planting type. Only irMPK4 plants in 75% irMPK4 populations had marginally higher assimilation rates, and EV plants in 75% irMPK4 populations had a higher WUE (**Figure 5C**, GLS, irMPK4 in 75% irMPK4: *emmeans* AMF-$_{(n = 3)}$ - AMF+$_{(n = 3)}$, t = −2.511, p=0.0363; GLS, EV in 75% irMPK4: *emmeans* AMF-$_{(n = 3)}$ - AMF+$_{(n = 3)}$, t = −8.148, p=0.0144). From these field and the previous glasshouse results, we conclude that the WUE phenotype is not likely to have accounted for the greater growth and yield of plants in low-irMPK4 populations.

## Shoot *MPK4* expression is required for *N. attenuata* to alter its reproductive yield when planted with a neighbor

We tested the effect of tissue-specific *MPK4* expression on plant yield responses to a neighbor. To separate the role of irMPK4 expression in shoots from those in roots, we created chimeric plants by micro-grafting EV shoots to irMPK4 roots (heterografts), EV shoots to EV roots (EV homografts) and irMPK4 shoots to irMPK4 roots (irMPK4 homografts; **Figure 6A**). Because the RNAi silencing signals travel from shoots-to-roots but not *vice versa* in *N. attenuata,* micrografting of RNAi lines such as irMPK4 does not permit the investigation of shoot-only *MPK4* knockdowns (**Fragoso et al., 2011**). Hetero- and homo-irMPK4 grafts retained similar levels of *MPK4* silencing in roots or roots and shoots, respectively (**Figure 2—figure supplement 1**). We grew the grafts under conditions of equal water availability, with or without an ungrafted EV neighbor. Photosynthetic parameter profiling of these grafted plants revealed that the heterografts were similar to EV homografts in assimilation, transpiration, stomatal conductance, and WUE (**Figure 6B**), while the irMPK4 homografts showed significantly higher transpiration rates, and stomatal conductance and lower WUE (**Figure 6B**; **Table 3**).

All graft types had shoot biomasses that were significantly reduced when plants were grown in pairs versus planted alone (**Figure 6C**; **Table 4**). The total root biomass per pot represented the roots of one plant for single pots, and two plants together for the paired pots. We compared the observed paired-pot root biomasses to a linear prediction of the paired-pot root biomass based on the addition of single-pot root biomasses of the respective graft types in the pair. Root biomasses of EV homograft pairs were equal to two times the root biomass of an EV homograft in a single pot. In contrast, the paired heterografts and irMPK4 homografts had smaller root biomasses than were predicted from individually grown plants (**Figure 6D**).

While both the EV homografts and heterografts displayed significant reductions in reproductive yield in response to an EV neighbor, irMPK4 homografts did not show a significant difference in

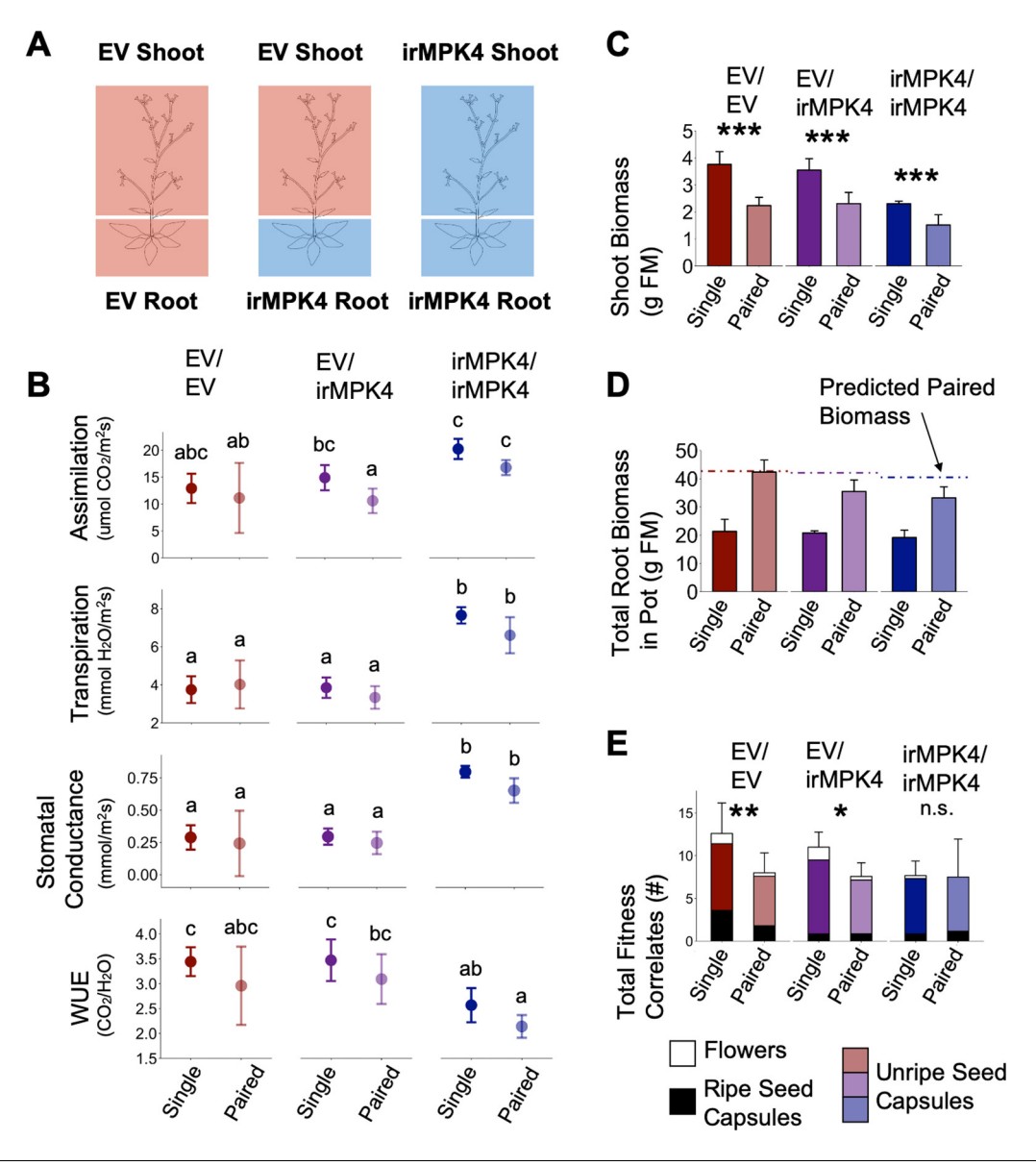

**Figure 6.** Expression of *MPK4* in the shoots mediates changes in *N. attenuata* reproductive output in response to neighbors. (A) EV shoots were micro-grafted onto irMPK4 roots, producing plants deficient in *MPK4* in the root but not in the shoot (EV/irMPK4, *Figure 2—figure supplement 1*). These were compared to EV/EV and irMPK4/irMPK4 homografts as controls. All three graft types were grown both as singles and in pairs with an ungrafted EV neighbor, under conditions of equal water availability in a glass house experiment (*Figure 6—figure supplement 1A*). (B) Assimilation rates, transpiration rates, stomatal conductance and water-use efficiency (WUE; mean ±CI, n = 3–7) of single and paired plants of each grafting type (EV/EV: red; EV/irMPK4: purple; irMPK4/irMPK4: blue) were measured at 37 dpp. Significant differences are presented across all graft and potting types. (C) Shoot biomass (mean + CI, n = 4–6) of EV/EV (red), EV/irMPK4 (purple) and irMPK4/irMPK4 (blue) individuals in each potting type (Single and Paired) was recorded at 50 dpp. Significant differences within genotypes are indicated (***: $p<0.001$). (D) Total root biomass in each pot (mean + CI, n = 4–6) was recorded for each potting type at 50 dpp. Dashed lines indicate the total pot root biomass predicted from the summed root biomasses of the respective genotype + EV/EV when planted as single plants in pots. (E) Counts (#) of fitness correlates (mean + CI, n = 5–7) of EV/EV (red), EV/irMPK4 (purple) and irMPK4/irMPK4 (blue) individuals in each pot type were recorded at 50 dpp. Statistical analyses were only performed for the total fitness correlates, although each bar is dissected into its contributing parts: flowers (white), unripe seed capsules (color), and ripe seed capsules (black). Significant differences within genotypes are indicated (*: $p<0.05$; **: $p<0.01$).

*Figure 6 continued on next page*

*Figure 6 continued*

The online version of this article includes the following source data and figure supplement(s) for figure 6:

**Source data 1.** Source data for *Figure 6*.
**Figure supplement 1.** Watering regime of the glasshouse grafted pair experiment.
**Figure supplement 1—source data 1.** Source data for *Figure 6—figure supplement 1*.

reproductive yield between single and paired pl ants (*Figure 6E*; *Table 4*). From these results, we conclude that silencing *MPK4* in the roots changes the neighbor-related root biomass production of *N. attenuata*, but *MPK4* in the shoots is required to alter reproductive yield in response to neighbors.

## Association with an AMF network abolishes biomass, but not reproductive overyielding in field populations with low percentages of *MPK4*-deficient plants

In order to evaluate if altering belowground interactions affects overyielding in 25% irMPK4 field populations, we compared the growth and yield of EV and MPxEV crosses, having the ability to interact with an AMF network, in field populations with varying percentages of MPxEV (0, 25, 75, 100%) with responses observed in populations with abrogated AMF interactions (*Figure 3*). Over-yielding was observed in unripe and ripe seed capsule counts in 25% MPxEV populations (*Figure 7D–E*), but not in the shoot and root biomasses RYTs for these same populations (*Figure 7B–C*). The overyielding in capsules, similar to the response in populations without AMF network associations, occurred as a result of increases in the number of capsules in EV plants, relative to the predicted yield based on their productions in monoculture.

To compare the biomass-to-reproductive-yield associations across populations with and without AMF association, we analyzed the data as allometric trajectories (*Weiner, 2004*; *Wu et al., 2003*). The presence of the AMF network significantly changed the allometric trajectories of EV individuals in 0% irMPK4 populations: EV plants had a significantly larger allocation to seed capsules per unit biomass than did EVxCC plants (*Figure 7F*, slopes: EVxCC$_{(n = 10)}$=0.73, EV$_{(n = 7)}$=3.3), and their

**Table 3.** Statistical *emmeans* contrasts within planting treatments for *Figure 6B**.

| Parameter | Model | Contrast | T value | P value |
|---|---|---|---|---|
| Assimilation | LM | S: irMPK4/irMPK4$_{(n = 6)}$ to EV/EV$_{(n = 5)}$ | −5.718 | 0.0001† |
| | | S: irMPK4/irMPK4$_{(n = 6)}$ to EV/irMPK4$_{(n = 7)}$ | −4.537 | 0.0014‡ |
| | | P: irMPK4/irMPK4$_{(n = 5)}$ to EV/EV$_{(n = 3)}$ | −3.666 | 0.0127§ |
| | | P: irMPK4/irMPK4$_{(n = 5)}$ to EV/irMPK4$_{(n = 6)}$ | −4.837 | 0.0007† |
| Transpiration | LM | S: irMPK4/irMPK4$_{(n = 5)}$ to EV/EV$_{(n = 5)}$ | −10.979 | <0.0001† |
| | | S: irMPK4/irMPK4$_{(n = 5)}$ to EV/irMPK4$_{(n = 6)}$ | −11.163 | <0.0001† |
| | | P: irMPK4/irMPK4$_{(n = 4)}$ to EV/EV$_{(n = 4)}$ | −6.506 | <0.0001† |
| | | P: irMPK4/irMPK4$_{(n = 4)}$ to EV/irMPK4$_{(n = 6)}$ | −9.008 | <0.0001† |
| Stomatal | LM | S: irMPK4/irMPK4$_{(n = 4)}$ to EV/EV$_{(n = 5)}$ | −9.429 | <0.0001† |
| conductance | | S: irMPK4/irMPK4$_{(n = 4)}$ to EV/irMPK4$_{(n = 7)}$ | −9.971 | <0.0001† |
| | | P: irMPK4/irMPK4$_{(n = 6)}$ to EV/EV$_{(n = 3)}$ | −7.209 | <0.0001† |
| | | P: irMPK4/irMPK4$_{(n = 6)}$ to EV/irMPK4$_{(n = 7)}$−9.079 < 0.0001*** | | |
| WUE | LM | S: irMPK4/irMPK4$_{(n = 6)}$ to EV/EV$_{(n = 4)}$ | 3.696 | 0.0109‡ |
| | | S: irMPK4/irMPK4$_{(n = 6)}$ to EV/irMPK4$_{(n = 7)}$ | 4.000 | 0.0051‡ |
| | | P: irMPK4/irMPK4$_{(n = 6)}$ to EV/EV$_{(n = 5)}$ | 3.240 | 0.0329§ |
| | | P: irMPK4/irMPK4$_{(n = 6)}$ to EV/irMPK4$_{(n = 6)}$ | 4.376 | 0.0019‡ |

*extracted from linear (LM) or generalized least squares (GLS) models with significant ANOVA results S: Singles; P: Paired.
†p value < 0.001; ‡p value < 0.01; §p value < 0.05.

**Table 4.** Statistical *emmeans* contrasts within planting treatments for **Figure 6C,E**[*].

| Parameter | Model | Contrast | T value | P value |
|---|---|---|---|---|
| Shoot Biomass | LM | EV/EV: $S_{(n = 5)}$ - $P_{(n = 5)}$ | −7.823 | <0.0001† |
| | | EV/irMPK4: $S_{(n = 4)}$ - $P_{(n = 6)}$ | −6.232 | <0.0001† |
| | | irMPK4/irMPK4: $S_{(n = 6)}$ - $P_{(n = 6)}$ | −4.442 | <0.0001† |
| TFC | LM | EV/EV: $S_{(n = 9)}$ - $P_{(n = 11)}$ | −2.637 | 0.0106‡ |
| | | EV/irMPK4: $S_{(n = 13)}$ - $P_{(n = 12)}$ | −3.620 | 0.0006† |
| | | irMPK4/irMPK4: $S_{(n = 13)}$ - $P_{(n = 10)}$ | −0.024 | 0.9813 |

*extracted from linear (LM) or generalized least squares (GLS) models with significant ANOVA results TFC: Total Fitness Correlates; S: Singles; P: Paired.
†p value < 0.001; ‡p value < 0.05.

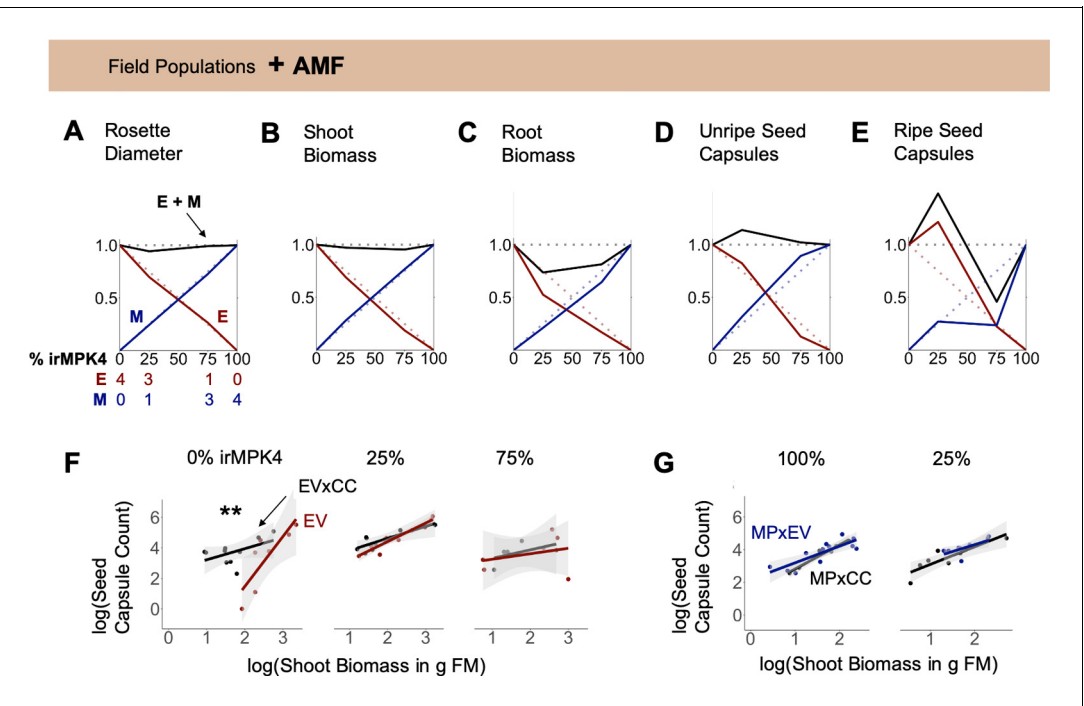

**Figure 7.** Interaction with arbuscular mycorrhizal fungi (AMF) abolishes overyielding in biomass, but not seed capsule production of populations with low percentages of *MPK4*-deficient plants. (**A – E**) EV and irMPK4 individuals crossed with EV instead of irCCaMK (EVxEV: EV; irMPK4xEV: MPxEV) can associate with arbuscular mycorrhizal fungi. Field populations were varied in percentages of EV and MPxEV plants (**Figure 3A**, **Figure 3— figure supplement 1**). Replacement diagrams show (**A**) rosette diameters (n = 12–28); (**B**) shoot biomasses (n = 6–12 excl. M in 75% = 2); (**C**) root biomasses (n = 7–12 excl. M in 75% = 2); (**D**) unripe seed capsules (n = 6–14 excl. M in 75% = 2); (**E**) ripe seed capsules (n = 6–16) of EV (E, red) and MPxEV (M, blue) plants in 0–100% irMPK4 field populations. Relative growth and yield for each genotype is calculated as (trait mean in mixture*# of plants)/(trait mean in monoculture*4). Relative yield totals (RYT, black) are calculated as E + M. Dotted lines indicate no deviations from yields in monocultures. (**F – G**) Allometric trajectories of (**F**) EV x irCCaMK (EVxCC, black) plants compared to EV (red) plants in 0%, 25% and 75% irMPK4 populations, as well as (**G**) irMPK4 x irCCaMK (MPxCC, black) plants compared to irMPK4xEV (MPxEV, blue) plants in 100% and 25% irMPK4 populations. Asterisks indicate significant differences within population types (*lstrends*, pairwise comparisons of slopes of fitted lines: ***p<0.001, **p<0.01, *p<0.05).

The online version of this article includes the following source data for figure 7:

**Source data 1.** Source Data for **Figure 7**.

trajectories started at a higher biomass threshold. In addition to the slope of the allometric trajectory, which indicates plasticity in resource allocation, the $R^2$ value, which indicates the extent to which a plant's trajectory is close to its reproductive potential (*Weiner, 2004*), also increased from EVxCC to EV plants in 0% irMPK4 populations (*Figure 7F*; $R^2$: EVxCC$_{(n = 10)}$=0.51, EV$_{(n = 7)}$=0.82). However, the allometric allocations of EV and EVxCC plants did not differ in 25% and 75% irMPK4 populations. MPxEV and MPxCC allometric trends did not differ in the 25% and 100% irMPK4 population type (*Figure 7G*). The 75% populations were excluded due to a lack of replication at the end of the field season.

We conclude that the loss of biomass overyielding in low-irMPK4 populations with AMF association is due to a change in the allometric trajectory of EV plants to higher biomass levels, which dwarfed EV biomass production in all other populations, without altering seed capsule production relative to the other populations. These results indicate that the overyielding does not require AMF-mediated belowground interactions.

## Discussion

Individual-level variation in resource-use traits, such as water-use efficiency (WUE), can change overall population yields (*Campitelli et al., 2016*; *Kenney et al., 2014*; *Montazeaud et al., 2017*). This occurs when plants respond to neighbors which are divergent in WUE with changes in growth and reproductive yield. However, the scale within a population's hierarchical organization (*Allen and Starr, 1982*) at which this phenomenon occurs is not yet known: yield changes in responding individuals (RIs) may be triggered only in immediate neighbors (neighbor scale; *Figure 1*) or in individuals across a population (population scale). Additionally, interactions may occur at these scales above- or belowground. Our analyses revealed that low abundances of irMPK4 plants intermixed with EV plants result in higher yields for *N. attenuata* populations, both in the glasshouse and the field (*Figure 3*). This overyielding effect was not caused by differences in soil water availability, which was controlled for in the glasshouse (*Figure 3H–M*; *Figure 3—figure supplement 5B* and *Figure 3—figure supplement 8*; *Figure 4*), nor irMPK4's WUE phenotype (*Figure 2A*), which was not different from the WUE of EV plants in the field (*Figure 5*). Interestingly, we find that yield-increasing responses in low-irMPK4 populations likely occurred aboveground, at the population scale (*Figures 4*, *5* and *7*).

Given that manipulating mixing proportions (*Weiner, 1980*) and total density (*He et al., 2005*; *Stachová et al., 2013*) are known to change overyielding results in substitutive experiments, it was important to consider that our glasshouse populations (12 plants, 5 cm apart) differed in plant density from our field populations (four plants, 10 cm apart). However, given that the overyielding results were consistent between the glasshouse and field, we infer that in our case the overyielding observed in mixtures with low proportions of irMPK4 was not due to differences in total plant density of mixtures.

Based on previous neighbor-effect studies using low-WUE phenotypes, we initially hypothesized that the neighbor scale (*Figure 1*, black arrow) would be critical for producing individual changes in yield. For example, in a paired-plant competition experiment, an *A. thaliana* CVI mutant (low WUE) produced more seeds than L*er* (high WUE) when both were planted with a control neighbor (*Campitelli et al., 2016*). When extrapolated to a population level, this result could predict a neighbor-scale change on population yield. Specifically, if one plant in each paired interaction were an RI, changes in populations yields would primarily occur when abundant RIs exist in the population in close proximity to few differentiated plants (i.e. populations with low percentages of well-spread differentiated individuals), creating many paired interactions without reducing RI numbers. Alternatively, if both plants were RIs, changed yields would be observed in populations with the either the highest abundance of interaction fronts between the two genotypes if the responses were in the same direction, or the lowest if they were in opposite directions. However, these possibilities rely on the assumption that a yield response is observed in response to trait variation in pairs. We therefore tested this pre-condition with EV and irMPK4 plants planted in monoculture and mixed culture pairs. Despite previously observed changes in neighbor responses to plants altered in their WUE (*Campitelli et al., 2016*), we did not see differences in any growth or yield parameters in mixed EV and irMPK4 pairs in comparison to monoculture pairs (*Figure 4*).

This discrepancy may be due to the difference between our methods and those of the earlier study: we controlled for water availability in our glasshouse experiments, to exclude that differences in water availability are driving the observed responses. Plants with low WUE can change the soil water availability in certain microenvironments and thus responses of neighbors due to the decreasing water table (*Zea-Cabrera et al., 2006*). However, we did not observe differences in soil moisture among field populations with different irMPK4 abundances (*Figure 3—figure supplement 4*). Even an additional watering treatment on a section of the field plot did not cause differentiated rates of drying-down among the different population types (*Figure 3—figure supplement 4*). We directly tested the influence of water availability in the glasshouse by controlling for soil water availability among populations (*Figure 3H–M*). We hypothesized that, with equalized soil water availability, we would not observe the overyielding response in low-irMPK4 populations. Interestingly, we again observed overyielding in low-irMPK4 glasshouse populations and therefore inferred the effect to be independent of soil water availability. Additionally, since controlling for soil water availability seemed to manipulate immediate neighbor plant interactions (i.e. differences between our results and *Campitelli et al., 2016*), the overyielding in controlled water glasshouse populations suggested that the effect was likely not mediated at the neighbor scale.

Paired-plant-in-a-pot experiments have only limited potential to study neighbor-scale interactions in population as individuals can only be observed in 0%, 50% and 100% irMPK4 mixtures (*Figure 4*). However, the growth and yield responses of plants at the neighbor scale could also depend on the identities of several immediate neighbors, or different percentages of immediate neighbors. Therefore, using a glasshouse population experiment, we compared the growth and yield of EV plants among a range of immediate irMPK4 neighbors (0 irMPK4 in 0% irMPK4 populations, 2 in 17% populations, and 4 in both 50% and 83% populations). We did not observe a correlation between the growth and yield of EV plants and the number of immediate irMPK4 neighbor plants (*Figure 3—figure supplement 6*). Based on these results and the lack of yield effects from our paired-pot experiments, we reject the hypothesis that the overyielding effects occurred at the neighbor scale, which suggest that the responses resulting in the observed population overyielding likely appear at the population scale.

At the population scale, we initially investigated known mechanisms through which individual plants could affect the yield generation of other individuals in the population, independent of changes in soil water availability. Trees with differing WUE influence photosynthetic parameters of neighboring trees (*Bunce et al., 1977*) and altered photosynthetic parameters of plants, including WUE, have been shown to cause significant yield changes in plants (*Hatfield and Dold, 2019*; *South et al., 2019*). Therefore, we examined the photosynthetic parameters of plants in the field and glasshouse populations. In the field, EV and irMPK4 plants did not differ in their photosynthetic parameters, including WUE (*Figure 5C*). In the glasshouse populations, irMPK4 showed a decreased WUE phenotype, but the photosynthetic parameters of EV and irMPK4 individuals in low-irMPK4 populations were not different from those in their respective monocultures (*Figure 5B*). From these results, we infer that the population-scale factor responsible for the low-irMPK4 population overyielding is independent not only of WUE, but also of the other previously observed photosynthetic phenotypes of irMPK4 plants, such as higher stomatal conductance, transpiration, and photosynthesis (*Hettenhausen et al., 2012*; *Hettenhausen et al., 2013*, *Figure 5B*).

Other factors that could cause the observed population-scale overyielding in low-irMPK4 populations include niche complementarity and the exchange of chemical signals. Hydrological niche partitioning could explain the increased population yield through a more diverse, and therefore more efficient, use of local water resources, either spatially or temporally (reviewed in *Barry et al., 2019*). In these instances, complementary plants may vary in above- or belowground water use through differences in water loss (e.g. transpiration, WUE; reviewed in *Silvertown et al., 2015*) or water acquisition (e.g. through root spatial distribution, *Dimitrakopoulos and Schmid, 2004*). However, the lack of differences in the WUE of irMPK4 plants and in the soil moisture among the different populations in the field, where overyielding effects were strongest (*Figure 3—figure supplement 4*), renders hydrological niche partitioning an unlikely explanation.

Even so, other types of complementarity could have occurred, for example through one neighbor type ameliorating the habitat of its neighbors. This typically occurs through abiotic means (i.e. changes in water or nutrient access; *Bertness and Callaway, 1994*; *Wright et al., 2017*) but again, the field results showed no differences in soil moisture or nutrient content among the soils of

different population types (*Figure 3—figure supplements 3* and *4*). However, the habitat could also be improved biotically: it is known that the presence of divergent individuals in a population that are resistant to a particular plant disease can reduce the spread of the disease, therefore allowing control plants in the population to produce more yield than when in monoculture (*Schmid, 1994*; *Zeller et al., 2012*). Although irMPK4 plants were shown to be more susceptible to the bacterial pathogen *Pseudomonas syringae* pv *tomato* (*Hettenhausen et al., 2012*), having the presence of susceptible individuals could still confer benefits to a population by serving as disease sinks (see e.g. *Keesing et al., 2006*). Alternatively, aboveground spatial complementarity could have occurred, where irMPK4 plants' smaller size (*Hettenhausen et al., 2012*) may have led to less competition for aerial space with EV plants, leading to their increased growth and population yield benefits (see e.g. *Lorentzen et al., 2008*; *Williams et al., 2017*). Although possible biotic factors could not be tested in the glasshouse, the role of spatial complementarity could be evaluated through a better understanding of the neighbor responses of EV and irMPK4 plants.

The response of a plant to a neighbor is commonly evaluated through comparisons of the plant's growth and yield when planted alone and when planted in a pair (*Díaz-Sierra et al., 2017*). We therefore included single plants in our paired plant experiments (*Figures 4* and *6*). Interestingly, EV plants were the only individuals that responded with decreases in growth and yield to the presence of a neighbor (*Figure 4*). Although the yield in paired-plant experiments cannot be directly compared to the yield of EV plants in various population types, it suggests that EV plants are RIs: they change their growth and yield in response to neighbors, unlike irMPK4 plants (*Figure 4*). It is conceivable that EV plants could benefit from only certain percentages of irMPK4 plant neighbors to cause an emergent effect at the population scale if irMPK4's consistent yield with or without a neighbor indicated a non-competitive response, though this cannot be distinguished in this study. However, complementarity effects would generally be expected to be strongest in equal mixtures, decreasing the likelihood of this possibility.

Volatiles can accumulate differently in the headspaces of various plant populations (*Schuman et al., 2015*) and have been shown to be a mechanism by which plants detect and respond to potentially competitive neighbors (*Engelberth and Engelberth, 2019*; *Ninkovic et al., 2016*; *Pierik et al., 2013*). irMPK4 and EV plants are known to emit distinct volatile profiles in response to herbivory: irMPK4 plants release 5x higher levels of *trans*-α-bergamotene than EV plants (*Hettenhausen et al., 2012*). As irMPK4 plants are inhibited in stomatal closure (*Hettenhausen et al., 2012*), they may continuously release a volatile or volatile blend that causes a population-scale effect only at a certain concentration (e.g. one that is generated in low-irMPK4 populations). This effect may be caused by a direct response of RIs, or might occur through interactions of the plant with various trophic levels, of which the latter has been observed to be volatile-mediated under field conditions (*Joo et al., 2018*). In *N. attenuata*, a population yield effect was previously shown to be affected by single gene variation through a tri-trophic interaction: plants silenced in one component of the jasmonic-acid defense pathway (i.e. defense-deficient plants) intermixed with wild-type controls increased the susceptibility of entire populations to a generalist herbivore, which then reduced total canopy damage by a successive specialist herbivore, and increased wild-type yield in mixed versus monoculture populations (*Adam et al., 2018*). *NaMPK4* is known to silence part of a jasmonic acid defense pathway; the uninhibited pathway in irMPK4 plants reduces the mass of a specialist caterpillar feeding on this line (*Hettenhausen et al., 2013*). This reduction of mass is often associated with higher mortality in young larvae due to disrupted feeding (*Cambron et al., 2019*; *McGale et al., 2018*), which could then reduce the presence of this specialist herbivore in the population, therefore decreasing canopy damage, and increasing the yield of control plants. Different volatiles released from irMPK4 plants could enhance this sort of effect (i.e. increase oviposition of this herbivore on irMPK4 plants due to increased attractive volatile emissions; *Kessler et al., 2008*) or otherwise might facilitate alternate multi-trophic interactions.

Belowground, root exudates can accumulate in populations and provide information for plants about their neighbors' identity and performance, which may result in plant growth responses (*Semchenko et al., 2018*). Root exudates can also promote certain microbial community in the rhizosphere that can confer benefits to the entire population (*Berg and Smalla, 2009*; *Huang et al., 2014*; *Fincheira and Quiroz, 2018*). However, irMPK4 root exudates have not yet been characterized. Further research on the chemical profiles and multi-trophic interactions of irMPK4 and EV plants in field populations is needed to identify potential above- or belowground factors which could

cause responses at the population scale, but perhaps first narrowing the effect to either the above- or belowground scale would be helpful to study these potential mechanisms.

We sought to distinguish the influence of the above- or belowground scale on the observed over-yielding by experimentally altering arbuscular mycorrhizal fungal (AMF) associations in field populations with varied amounts of *MPK4*-deficient plants. Although AMF associations have a strong impact on plants' access to soil water and nutrients (*Egerton-Warburton et al., 2007*; *Reynolds et al., 2003*; *Yang et al., 2013*), as well as on belowground plant-plant interactions (*Ferlian et al., 2018*; *Gorzelak et al., 2015*; *Song et al., 2019*), we consistently observed reproductive overyielding in low-irMPK4 populations regardless of the AMF association (*Figure 3*; *Figure 7*). The lack of change in the reproductive overyielding pattern supports further attention to above-ground tissues for mechanistic determination, although it does not exclude other belowground inter-actions as potential causative factors.

To investigate the tissue dependence of the EV and irMPK4 plant neighbor responses we had previously studied in our paired experiment, we again used a single versus paired experimental set-up with grafted plants. The grafting experiment revealed that *MPK4* transcript accumulation, or an MPK4-dependent downstream factor, is required in aboveground tissues for *N. attenuata* plants to be able to change their yield in response to neighbor presence (*Figures 4F* and *6E*). Additionally, the presence of *MPK4* transcripts in the shoot alone is sufficient to facilitate the yield-altering neigh-bor response, specifically in reproductive correlate production (*Figure 7E*). This tissue-specific role of *MPK4* abundance in the shoot could be important in the mediation of reproductive overyielding in populations, but would need to be tested in a population experiment. Because the RNAi silencing signals travel from shoots to roots in *N. attenuata* (*Fragoso et al., 2011*), we were unable to investi-gate the function of shoot-only *MPK4* knockdowns. The use of plants with mutant or natural alleles of *MPK4* or homologs, such as those identified *in A. thaliana* (*Des Marais et al., 2014*), would allow for the analysis of reciprocal *mpk12/wt* and *wt/mpk12* grafts. Together with a metabolic characteri-zation (e.g. volatiles) and evaluations of multi-trophic interactions, these grafting experiments could identify the mechanisms mediating the observed population scale overyielding.

The results of our study are consistent with the well-established phenomenon that divergent indi-viduals in a community can increase community productivity (*Chapin III, 1997*; *Chapin, et al., 1998*; *Crutsinger, 2016*; *Hooper et al., 2005*; *Naeem et al., 1994*; *Schulze and Mooney, 1994*) and advance our understanding by identifying the spatial scale (neighbors vs. population, above vs. belowground) at which this yield response occurs. Additionally, we identify a single gene, *MPK4*, which when silenced in low abundances in a population, is responsible for population overyielding. By excluding neighbor-scale effects and controlling resource availability, we demonstrate that *MPK4* influences population yield through RIs at the population scale independent of water availability, but further experiments are needed to identify the specific mechanisms mediating this effect. This work contributes to our understanding of how populations may become more productive as a result of greater genetic and functional diversity and suggests that experiments exploring the scales at which these effects occur can identify novel means to increase the productivity of agronomic monocultures.

# Materials and methods

**Key resources table**

| Reagent type (species) or resource | Designation | Source or reference | Identifiers | Additional information |
|---|---|---|---|---|
| Genetic reagent (*N. attenuata*) | A-04-266-3 | *Bubner et al., 2006* DOI: 10.1007/s00299-005-0111-4 | | Empty vector control |
| Genetic reagent (*N. attenuata*) | A-7–163 | *Hettenhausen et al., 2012* DOI: 10.1086/342519 | | Stably silenced in *MPK4* via RNAi |
| Genetic reagent (*N. attenuata*) | A-09-1212-1-4 | *Groten et al., 2015* DOI: 10.1111/pce.12561 | | Stably silenced in *CCaMK* via RNAi |
| Software | R version 3.4.2 | *R Development Core Team, 2017* | | |

*Continued on next page*

*Continued*

| Reagent type (species) or resource | Designation | Source or reference | Identifiers | Additional information |
|---|---|---|---|---|
| Software | RStudio version 1.0.153 | *Rstudio Team, 2016* | | |

## Plant material and constructs

Characterization of the empty-vector (EV) *Nicotiana attenuata* control line (pSOL3NC, line number A-04-266-3) is described in *Bubner et al. (2006)*. The irMPK4 line (pRESC5MPK4, line number A-7–163), silenced in the production of MITOGEN-ACTIVATED PROTEIN KINASE 4 (MPK4) through RNAi targeting *MPK4* transcripts, is characterized in *Hettenhausen et al. (2012)*; *Hettenhausen et al. (2013)*. The irCCaMK line (pSOL8CCAMK, line number A-09-1212-1-4), silenced in the production of CALCIUM AND CALMODULIN-DEPENDENT PROTEIN KINASE (CCaMK) through RNAi targeting *CCaMK* transcripts, is characterized in *Groten et al. (2015)*.

EVxirCCaMK (pSOL3NCxpSOL8CCAMK, 'EVxCC') and irMPK4xirCCaMK (pRESC5MPK4xpSOL8CCAMK, 'MPxCC') crosses were generated by growing homozygous EV (second generation, T2), irMPK4 (T2) and irCCaMK (third generation, T3) in the glasshouse and hand pollinating the styles of EV and irMPK4 emasculated flowers with pollen from the anthers of irCCaMK flowers. Control crosses EVxEV (pSOL3NCxpSOL3NC, 'EVxEV') and irMPK4xEV (pRESC5MPK4xpSOL3NC, 'MPxEV') with the same paternal genotypes were created by pollination with pollen from EV. Hand-pollinated flowers were tagged with string and resulting seed capsules were collected. The ripe seeds from these crosses provided the seed source for the field population experiment (*Figure 3*). A characterization experiment in the glasshouse revealed that EVxEV and EVxCC, as well as MPxEV and MPxCC, were not significantly different in water loss rates per day (*Figure 3—figure supplement 7A*, *Supplementary file 1*), shoot and root biomass (*Figure 3—figure supplement 7B–C*). For all other experiments in the glasshouse, $T_3$ generation EV and irMPK4 homozygous lines were used.

## Plant growth conditions

Importation and release of transgenic crosses in the field station (Lytle Ranch, UT) was carried out under Animal and Plant Health Inspection Service (APHIS) import permit numbers 07-341-101n (EV) and 10-349-101m (EVxirCCaMK, irMPK4xirCCaMK, irMPK4xEV), and release 16-013-102r.

Glasshouse and field germination and growth were described previously (*McGale et al., 2018*), with modifications only in planting design. Field plants were planted in four-plant populations in a square design (*Figure 3*; *Figure 3—figure supplement 1*), with 10 cm between each adjacent neighbor. Plants of the glasshouse population experiment were potted in 12-plant populations (*Figure 3*; *Figure 3—figure supplement 5*), with 5 cm between each adjacent neighbor. Glasshouse plants in both of the paired experiments (grafted and ungrafted) were also planted 5 cm from their neighbor plants. The planting substrate consisted of a bottom layer of large clay aggregate (Lecaton, 8–16 mm diameter, approximately 10% of pot volume), a central layer of small clay aggregate (Lecaton, 2–4 mm diameter, approximately 80% of pot volume) and a top layer of fine sand (approximately 10% of pot volume). This substrate provides optimal drainage in the pots for the purposes of water control, and conditions similar to the sandy, clay soil of the natural habitat of *N. attenuata*.

## Plant growth and yield measurements

For the field experiment (*Figure 3A–G*), rosette diameter measurements were extracted from photos taken between 19:00 and 20:00, in which each individual plant was pictured next to a standard metal square (5 × 5 cm) for scale. Plant stalk height measurements were recorded as the height from the base of the stalk at the ground level to the highest point of the topmost inflorescence. Plant shoot and root dry biomass were measured by placing respective biological matter in a paper bag inside of a plastic box with ventilation holes of 1 cm diameter drilled through the lid and left to dry for 15 days in the sun, before being removed from the bag and weighed. Unripe seed capsules were counted simultaneously for all plants, immediately before harvesting for shoot and root biomass. Due to APHIS regulations, ripening seed capsules were counted and subsequently removed to prevent opening and releasing seeds into the field; the total ripe capsules collected is presented (*Figure 3G*).

For all glasshouse experiments (*Figures 3H–M*, *4* and *6*), rosette diameter was measured directly on the plant. Plant stalk height was measured as in the field. Shoot biomass consisted of all above-ground matter (severed below the rosette), placed inside a bag for drying at 80℃ for 2 hr, after which the plant matter was removed from the bag and weighed. The shoot biomass was also weighed for fresh mass, and the water content of the plant at harvest was reported as the difference between the fresh and dry shoot biomasses. All fitness correlates were counted at harvest, including flowers (counted as flowers when the corolla became visible by pushing through the sepals), unripe and ripe seed capsules, and the total of all of these together was reported (*Figure 3M*).

## Soil moisture and element content

Soil cores were taken from the field by driving a split tube core borer (53 mm, Eijkelkamp, Giesbeek, Netherlands) 30 cm into the ground, and carefully removing it with the core intact. 5 cm pieces of field soil were cut from the core from 0 to 5, 10 to 15, and 25 to 30 cm below ground. Each of these 5-cm-thick sections were weighed, left to dry in the sun in UV-excluding boxes similar to those used for the drying of shoot biomass (see *Plant growth and yield measurements*), and weighed again when dry (determined to be when the mass fluctuated <0.1 g between days). Soil moisture was calculated for each sample (% soil moisture = (fresh soil mass - dry soil mass/fresh soil mass) * 100), taken from 21 to 30 dpp in the different population types (*Figure 3—figure supplement 4*, n = 1 per population).

Soil cores were obtained using the same method at 54 dpp with replication (n = 2–9) to determine the soil content of total, inorganic and organic carbon ($C_{total}$, $C_{inorg}$, $C_{org}$, respectively), nitrogen (N), copper (Cu), iron (Fe), potassium (K), phosphorus (P), and zinc (Zn) in each type of population at the end of the season (*Figure 3—figure supplement 3*). Samples were dried at 80℃ for 6 hr in a drying oven, sieved and milled for $C_{total}$ determination (elemental analyzer; High TOC, Elementar, Hanau, Germany), $C_{inorg}$ (loss-on-ignition from elemental analyzer), $C_{org}$ ($C_{total}$ - $C_{inorg}$), and N (elemental analyzer) at the Max Planck for Biogeochemistry in Jena, Germany. Cu/Fe/K/P/Zn concentration were determined by microwave digestion and atomic absorption spectroscopy (*Karpiuk et al., 2016*).

## Water treatments

Field populations were watered every week for 1 hr at dusk (20:00 to 21:00) from a central water dripper (2 L/h drip rate) present in each population. After 34 dpp, one section of the plot was no longer watered until the final harvest (Dry), while a small subsection was watered two more times (Wet) in order to obtain gas exchange measurements on sections with varying water treatments at 48 dpp. Soil moistures at 21–30 dpp in these two parts of the plot (see *Soil moisture and element content*) were analyzed by regression to test if results from both of these parts could be summarized together in *Figure 3* (*Figure 3—figure supplement 4*). Watering treatment and the interaction with depth or day did not significantly predict soil moisture (*Figure 3—figure supplement 4B*, Wet subsection: 'Part2'; day: 'variable'; model fit: $R^2$ = 0.406, F(7, 147)=16.04, p-value=1.474e-15). Therefore shoot and root biomass, as well as unripe and ripe seed capsule data collected from full populations in both sections were reported together as one mean (*Figure 3*).

In the glasshouse, all populations and pairs (grafted and ungrafted) underwent the following regimented watering to control for water availability: after potting, pot were given establishment watering (soil moisture maintained around 20%), allowing root development to the bottom of the pot for a transition from top watering to bottom watering. After 3 weeks, pots with population types began to show detectable differences in water loss and consumption-based watering began at ecologically relevant soil moistures (*Valim et al., 2019*). This reflected the known decrease in soil moisture throughout the life cycle of *N. attenuata* in the field (*Zavala and Baldwin, 2004*). For the population and pair experiment, ecologically relevant soil moisture was achieved by daily watering of individual pots to a 2-day water supply, calculated as:

$$WM = 2^*\text{mean}(WL_{-1}, WL_{-2}) + DP$$
$$WM = \text{pot mass (g) to which the pot needed to be watered}$$
$$WL_{-1} = \text{water loss (g) from the previous to the current day}$$
$$WL_{-2} = \text{water loss (g) from two days to one day prior}$$
$$DP = \text{dry pot mass (g of pot with dry substrate, before planting)}$$

The 2-day water supply is illustrated for our glasshouse paired experiment (*Figure 4B*). To allow larger growth and thus accentuate growth differences in plants in the grafted pair experiment (*Figure 6*), the water supply was raised to 5 days (WM = 5*mean(WL$_{-1}$, WL$_{-2}$) + DP), bringing soil moisture percentages up to 20–30%. The higher soil moisture did not affect the differences in photosynthetic parameters of EV and irMPK4 homografts compared to those reported for the homozygous EV and irMPK4 plants in the paired experiment (*Figures 5A* and *6B*). There was no significant correlation between the amount of water added in our watering regimes and the amount of water lost (demonstrated two times during watering regime of the grafted experiment, *Figure 6—figure supplement 1B–C*).

## Leaf turgor and potential effects of controlled watering on diurnal rhythms

Yara ZIM-probes were placed on 2–3 replicates of EV or irMPK4 plants in all glasshouse population types (*Figure 3H*). The probes consist of two magnets clipped onto both sides of a leaf, of which the lower magnet includes a pressure sensor. All probes are initialized at a clamping pressure between 10 and 30 kPa on a turgescent leaf. The probes allow continuous measurement of leaf turgor pressure throughout an experiment, and we present 48 hr of continuous monitoring from 00:00 December 4th to 00:00 December 7th (*Figure 3—figure supplement 8*).

Absolute pressure values could not be compared quantitatively; in contrast to previously reported measurements (*Zimmermann et al., 2008*) our recordings initialized at different clamping pressures. A comparison of three replicates of irMPK4 in 100% irMPK4 populations showed that variation in peak-trough amplitudes within one genotype/population type (irMPK4 in 100% irMPK4 populations; *Figure 3—figure supplement 8C*) exceeds variation between EV and irMPK4 in all other populations (*Figure 3—figure supplement 8A–B*). Therefore, we compared the time of day at which changes in turgor pressure values occurred. For all EV and irMPK4 plants in all population types, the diurnal turgor pressure changes (peaks and troughs) do not occur at different times (*Figure 3—figure supplement 8A–B*).

Additionally, we observed if our genotypes experience differential diurnal dry-downs or unexpected drought events that may not be captured by our daily pot weighing and watering for our controlled water treatment. This may be reflected in 'noisier' curves (increased oscillations within the peaks or troughs of the diurnal leaf turgor changes) or inverted leaf turgor pressure curves around noon (*Martínez-Gimeno et al., 2017*), however, we did not observe any of these qualities across our measured plants. We therefore inferred that our controlled watering treatment was not causing unknown diurnal drying differences among individuals in our glasshouse population types and proceeded with applying it to all glasshouse experiments (*Figure 3H–M*, *Figure 4*, *Figure 5A–B*, *Figure 6*).

## Gas exchange measurements and water-use efficiency calculations

Gas exchange measurements including photosynthesis and transpiration rates, and stomatal conductance (via calculation), were performed using a LI-COR 6400XT infrared gas analyzer (Lincoln, NE), both in the field and the glasshouse between 12:00 and 14:00 (*Figure 5*).

The LI-6400XT was combined with a Leaf Chamber Fluorometer in the glasshouse to additionally obtain chlorophyll fluorescence measurements after 6 hr of dark adaptation (lights off at 22:00, measurements from 4:00 to 6:00; *Figure 5—figure supplement 1A*). A saturating pulse of light was applied to the dark-adapted leaves to ensure that all photosystem II (PII) energy was released as fluorescence and detected as the $F_m$ value. $F_v$ was calculated from $F_m$ minus $F_0$ ($F_0$ being the base level of fluorescence emitted without the saturating pulse). $F_v/F_m$ represents the maximum quantum yield of PII, which was used as a measure of photosynthesis limitations unrelated to stomata (*Signarbieux and Feller, 2011*). Photosynthesis and transpiration rates were also acquired concomitantly with $F_v/F_m$ values during this pre-dawn sampling, and stomatal conductance and WUE were calculated (*Figure 5—figure supplement 1B–E*).

Water-use efficiency (WUE) was calculated as the ratio of photosynthetic rate ($\mu mol\ CO_2/m_2s$) to transpiration rate ($mmol\ H_2O/m_2s$), thus resulting in units of carbon dioxide molecules used per 1000 water molecules (*Figures 5A,B,C* and *6B*; *Figure 5—figure supplement 1E*).

## Micro-grafting

Seven-day-old seedlings were micro-grafted as described previously (*Fragoso et al., 2011*), with EV scions grafted to both EV (EV/EV) and irMPK4 (EV/irMPK4) rootstocks, and irMPK4 scions grafted only to irMPK4 (irMPK4/irMPK4) rootstocks (*Figure 6*). The average grafting success was 90% (p>0.05 between genotypes, ANOVA, Tukey HSD *post hoc*).

## Transcript abundance

RNA was extracted with TRIzol reagent (Invitrogen) according to the manufacturer's instructions. cDNA was synthesized from 500 ng of total RNA using RevertAid H Minus reverse transcriptase (Fermentas) and oligo (dT) primer (Fermentas). qPCR was performed in a Mx3005P PCR cycler (Stratagene) using 5X Takyon for Probe Assay (No ROX) Kit (Eurogentec), TaqMan primer pairs and double fluorescent dye-labeled probe. *N. attenuata Sulfite Reductase* (*ECI*) was used as a standard housekeeping gene, and its primer sequences and probe, as well as the *MPK4* primer sequences and probes, are as published previously (*Wu et al., 2007*). *MPK4* transcript levels were quantified relative to the housekeeping gene as described in Wu et al (*Figure 2C* and *Figure 2—figure supplement 1*; *Wu et al., 2007*).

## Statistical analysis

All data were analyzed using R version 3.4.2 (*R Development Core Team, 2017*) and RStudio version 1.0.153 (*Rstudio Team, 2016*). Replication for experiments is indicated in the figure captions. The replacement diagrams in *Figures 3*, *4* and *7* do not display statistical significance, but facilitate the visualization of cumulative population overyielding (*de Wit, 1960*). Statistical means of the data used to produce these diagrams are presented in *Figure 3—figure supplements 2* and *6*.

Some pseudoreplication resulted from plants being measured from within the same population or pot throughout our experiments (*Figures 3–4*, *7* and *Figure 3—figure supplements 2* and *6*). We evaluated whether this effect contributed significantly to changes in our dependent variable using ANOVA comparisons of nested linear mixed effects models (i.e. LME/R models with and without the pseudoreplication as a random effect) as described by Zuur et al. (R packages *lme4*, *nlme*; *Bates et al., 2015*; *Pinheiro and Bates, 2019*; *Zuur et al., 2009*). Pseudoreplication was only included as a random effect in the respective LME/R model if it was significant; the model was then fitted for its fixed effects, and was checked for outliers (through Cook's distance and leverage plots), homoscedasticity and normality (through graphical analysis of residuals; *Zuur et al., 2009*). Pairwise *post hoc* comparisons of fixed effects were extracted from the model using the R package *emmeans* (*Lenth et al., 2019*), following their significance in an ANOVA. ANCOVA analyses were not used as the variable representing pseudoreplication (i.e. population or pot number) is inherently non-independent, which violates the assumptions for testing the significance of a covariate with ANCOVA, but does not violate the assumptions of a random effect in a mixed effects model.

Datasets without significant pseudoreplication were fit to the best suited of either a linear model (LM; RC Team Package *stats*), a generalized least squares model (GLS, R package *nlme*) or an LME model, and were checked for outliers, homoscedasticity and normality (as above; *Zuur et al., 2009*). Pairwise *post hoc* comparisons for fixed effects were extracted as above, or with Tukey HSD tests (*R-core Team, 2015*) following their significance in ANOVA.

Regression analyses (*Figure 7F,G*) were performed using the *lstrends* function (R package *emmeans*) and statistical significances were extracted using *pairs* (*R-core Team, 2012*).

## Acknowledgements

We thank Brigham Young University for use of their awesome field station, the Lytle Ranch Preserve; APHIS for constructive regulatory oversight of the release of transgenic plants; the glasshouse team at the Max Planck Institute for Chemical Ecology and colleagues in the 2016 field work team for support the technical staff at the Department of Molecular Ecology, particularly Dr. Klaus Gase, as well as Lucas Cortes Llorca, for advice on genetic material extractions; the International Max Planck Research School (IMPRS) on the Exploration of Ecological Interactions with Chemical and Molecular Techniques and the Young Biodiversity Research Training Group - yDiv for their support of EM and HV; and the reviewers of this manuscript, whose advice was essential in revising the manuscript.

## Additional information

### Competing interests

Ian T Baldwin: Senior editor, *eLife*. The other authors declare that no competing interests exist.

### Funding

| Funder | Grant reference number | Author |
|---|---|---|
| Max-Planck-Gesellschaft | | Erica McGale<br>Henrique Valim<br>Deepika Mittal<br>Jesús Morales Jimenez<br>Rayko Halitschke<br>Meredith C Schuman<br>Ian T Baldwin |
| European Research Council | Advanced Grant 293926 | Henrique Valim<br>Meredith C Schuman<br>Ian T Baldwin |
| iDiv | | Erica McGale<br>Henrique Valim<br>Meredith C Schuman |

The funders had no role in study design, data collection and interpretation, or the decision to submit the work for publication.

### Author contributions

Erica McGale, Conceptualization, Data curation, Software, Formal analysis, Validation, Investigation, Visualization, Methodology, Project administration, Writing - original draft; Henrique Valim, Data curation, Software, Formal analysis, Investigation, Methodology; Deepika Mittal, Data curation, Investigation, Methodology; Jesús Morales Jimenez, Conceptualization, Data curation, Investigation, Methodology; Rayko Halitschke, Supervision, Writing - review and editing; Meredith C Schuman, Conceptualization, Supervision, Funding acquisition, Validation, Investigation, Methodology; Ian T Baldwin, Validation, Data curation, Investigation, Methodology, Project Administration, Writing - reviewing and editing

### Author ORCIDs

Erica McGale  https://orcid.org/0000-0002-5996-4213
Rayko Halitschke  http://orcid.org/0000-0002-1109-8782
Meredith C Schuman  https://orcid.org/0000-0003-3159-3534
Ian T Baldwin  https://orcid.org/0000-0001-5371-2974

### Decision letter and Author response

Decision letter https://doi.org/10.7554/eLife.53517.sa1
Author response https://doi.org/10.7554/eLife.53517.sa2

## Additional files

### Supplementary files

• Supplementary file 1. Table of *emmeans* contrasts, within genotypes from *Figure 3—figure supplement 6*[α].

• Transparent reporting form

### Data availability

The datasets used in the final fittings of our linear (LM), generalized least-squares (GLS), or linear mixed effects (LME/R) models, from which statistical significances were extracted, are included as

Source Data files in our submission. Each Source data file refers to the figure in which the corresponding data/results are displayed.

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
