## [Decision Letter]

**Acceptance summary:**

Plant populations can have increased yield if individuals differ in a single gene, but this effect may require a particular mixing ratio of the involved genotypes and thus cannot be achieved by stacking the gene variants into a single genotype or pairing genotypes simply at neighborhood scale. The authors of this study used sophisticated experiments to chase the phenotypic differences between the two genotypes causing the population-level overyielding. Interestingly, the most obvious phenotypic difference in water-use efficiency was not the cause, leaving above-ground resource partitioning or chemical signaling as most likely alternatives.

**Decision letter after peer review:**

[Editors’ note: the authors submitted for reconsideration following the decision after peer review. What follows is the decision letter after the first round of review.]

Thank you for submitting your work entitled "Determining the scale at which variation in WUE traits changes population yields" for consideration by *eLife*. Your article has been reviewed by three peer reviewers, including Bernhard Schmid as the Reviewing Editor and Reviewer #1, and the evaluation has been overseen by Christian Hardtke as Senior Editor.

Our decision has been reached after consultation between the reviewers. Based on these discussions and the individual reviews below, we regret to inform you that your work will not be considered further for publication in *eLife*.

The strength of you manuscript is the use of very promising tools to analyze a within-species diversity effect mechanistically down to the effects of a single genetic difference between members of a population. The weakness is the ecological setting of the work as detailed in the reviews below. This aspect could be improved by setting a different frame, but we also feel it would need different or additional experiments. All reviewers found your manuscript very difficult to understand, and I think some of the comments reflect this, for example those about statistics in the third review. It is possible that some of these comments are not entirely correct, but then it is very difficult to find out how you did the actual statistical analysis. I'm sorry for the negative result of this long review process. But I still hope that the reviews are helpful and that you continue to use your excellent tools and understanding of plant interactions at the molecular and chemical level to analyze population-level and community-level processes that have been described quite phenomenologically by ecologists.

*Reviewer #1:*

This is an interesting yet somewhat complicated manuscript. The interesting question is whether plant populations can show *emergen*t features that a single individual cannot express. For example, a mixture of varieties resistant to different mildew strains may lead to the population being protected from infection even when each variety grown by itself would be infected. Obviously, it would not be possible to stack all resistant genes into a single variety because of excessive fitness costs, but at the population level a solution is possible by mixing the different varieties. Another example would be that a deep-rooting variety could provide water to a shallow-rooting variety by hydraulic lift during drought and reciprocal benefits at other times. Mechanisms of how emergent features can be created by diversity have been discussed at large in the biodiversity-ecosystem functioning literature, but their molecular basis has almost never been assessed, except in a recent paper by Wuest and Niklaus (Nature Ecology and Evolution 2018).

In the present paper it is a mixture of *Nicotiana attenuata* genotypes with (EV) or without (irMPK4)-in simplified words-the ability to control water loss via their stomates that leads to an emergent feature at the population level, namely an increased population performance. The first idea is of course that the greater water uptake of irMPK4 would affect the water uptake at population level, but this possibility could be excluded using a watering regime that compensated for the water-wasting behaviour of this genotype. Thus, in the end we are left with a similar situation as in many biodiversity experiments: there are clear effects of *overyielding* in diverse populations or communities, but we cannot identify the mechanisms. Nevertheless, this paper offers a new view and approach in several ways (single gene difference between individuals, maintenance of constant water level, exclusion of mycorrhizae by using plants with irCCaMK background, micro-grafting of individuals), which can inform further research into diversity effects as well as how individuals within a population of a single species may show correlative or *coordinated* behaviour to produce emergent population-level features.

The authors focus in my view perhaps too much on a comparison between neighbor interactions and effects at neighborhood scale and population-level responses. This originates from the hypothesis that it could be chemical signals that are exchanged between individuals that lead to the observed phenomena. However, other, perhaps more phenomenological mechanisms would be conceivable and would typically be considered in biodiversity-ecosystem functioning research. These include resource complementarity or interactions with other organisms, including soil microbes. Basically, the question is why a few individuals that have lost the ability to control their stomates can increase population performance. I would recommend to consider it this way, i.e. EV plants are responsive and can for example increase their performance if planted individually instead of in pairs, as expected by the law of constant final yield, whereas irMPK4 plants are less responsive and cannot fully compensate for the reduced density when grown as individuals (see your Figure 5, cf. Stoll et al., PRSLB 2002, https://doi.org/10.1098/rspb.2002.2137).

*Reviewer #2:*

The manuscript presents the results of a large multistage project in which plant performance was studied at variable organizational scales in populations of *Nicotiana attenuata* comprising variable proportions of low-WUE individuals. The results of a field experiment and two greenhouse experiments suggest that the presence of a low proportion of low WUE individuals may increase population-level performance. The study touches on very interesting and rather neglected aspects of population-level consequences of variability in stress vulnerability and adaptions, and as such it is both timely and important. While many aspects of the work, especially those related to the verification of mutant functionality, seem to have been executed meticulously, the experimental design and collected data raise serious doubts about the interpretation and significance of the findings.

1) Water availability: studying the functional implications of the presence of low-WUE individuals calls for experiments in which water availability is deliberately varied, controlled and monitored. Such a design would contrast high-water conditions where the presence of low-WUE individuals would be expected to cause little population-level effects and low-water conditions under which the presence of such plants would be expected to trigger significant population-level effects. While the authors tried to create differential water treatments in the field experiment, this attempt did not succeed. Moreover, in the two greenhouse experiments differential water availability was not even attempted and instead the authors provided constant water supply by applying frequent water supplements. Besides missing the above-mentioned aspect of differential water availability, this protocol might or might not have sufficiently homogenized water availability for the plants. Even if soil water content seemed to have been equalized by the daily supplementation, soil and plants surely underwent daily cycles of diurnal drying (more so in populations with larger proportions of low WUE plants), which could have had important effects on plant drought perception, responses and population effects. Overcoming this problem could have been achieved by running at least two (high vs. low) water treatments that would be continuously sustained with high precision and temporal resolution (minutes, not whole days) and soil water content should have been monitored using non-destructive methods, e.g. TDR, to account for high-res variability in water availability. Accordingly, while the findings of these experiments are legitimate, their interpretation could be different. Specifically, it can be hypothesized that the observed population-level responses to the presence of low proportions of low-WUE plants was triggered by the slightly reduced water availability in those treatments (rendered by these plants) and not necessarily by any non-resource signaling or cuing. Teasing apart the resource availability and non-resource information alternatives would require additional treatments and much tighter monitoring of soil water (see above).

2) Neighbor- vs. population-level effects: based on careful analyses of the response of individuals located at variable distances from the irMPK4 plants, the authors claim that the population-level effects of the presence of low proportion irMPK4 plants could not be attributed to local interactions between immediate neighbors; however, given that a) the effect is most likely done via root-root interactions and b) the precise position of the roots in such micro populations is not known, the reported results could be hypothesized to have been caused by the very commonly observed aggregation of more active roots near the pot's boundary/wall, where most root interaction took place. Accordingly, the seemingly counter-intuitive 'scale-skipping' phenomenon could be trivially attributed to a technical artifact rather than to an ecologically meaningful phenomenon.

3) Abrogation of AMF associations: the hypothesis that mycorrhizal networks could play an important functional role in the priming of neighboring plants to various stresses has been supported in other cases and could have been interesting in the context of the present project. It is thus not clear why the authors chose to eliminate this aspect from the study.

4) Fitness effects: while proxies of fitness typically, and for very good reasons, serve as golden evidence in ecological studies, attention should be given to the fact that the harvests and fitness surveys were conducted at ca. day 50 and before plants had the opportunity to finish their natural life cycle (typically 90 d). Given that the presence of a low proportion of low-WUE plants could have slightly reduced water availability, the observed differences could be attributed to accelerated reproduction, which is rather common in plans responding mild stresses or signals and cues of imminent stresses (see e.g. https://doi.org/10.1093/jxb/erw272).

5) Shoot involvement in MPK4-deficiency: the demonstration of the importance of the shoots in mediating the functionality of MPK4-deficiency is very nice but not directly related to the main questions dealt with in the study. Given the vastness of the manuscript, it seems that this finding could be reported elsewhere.

6) Data analyses and presentation: a) Analysis of a large proportion of the results could have been better done using ANCOVAs, accounting for individual data, not merely treatment means.

b) The graphs are extremely cumbersome, complex and hard to follow. Lumping result by analysed plant and not paired by treatment is extremely hard to follow and non-intuitive. The tiny trend insets seem jargony as the order of the presented data is not intuitive (see previous comment).

c) Why CIs are not presented for population-level data?

d) The main text is heavy with statistical information that could be moved to tables or to figure legends.

e) In most cases, the presentation of means could be much clearer using colored points and CI bars rather than colored bars.

Additional points:

Introduction, first paragraph, subsection “MPK4 is necessary for *N. attenuata’s* growth and yield responses to neighbors”, last paragraph, and elsewhere: not necessarily 'information' as such as performance can be (simply) affected by water availability (see above comments).

Subsection “Plant growth and yield measurements”, second paragraph – 8-16mm,

Subsection “Water treatments”, last paragraph: it is unrealistic to translate field water availability to greenhouse conditions (even in very large pots).

Figure 4: incorrect plate numbering- B=A, C=B etc.

Figure 5: continuous X-Y graphs (linear time units on the X axis) would be more appropriate for time series.

Figure 5C: bar colors are too dark and thus not sufficiently distinguishable from each other.

*Reviewer #3:*

McGale et al. evaluate the role of a single gene, MPK4, on the population yield of *Nicotiana attenuata*. In general, the experiments in the study appear to have been well-designed and comprise an elegant set of tests in the search for a mechanistic explanation of the increase in yield.

Regarding the different methods, I did not follow the author's justification for using PATs. The authors calculate mean traits for each genotype in each treatment, and then calculate the predicted summed traits from genotype means and genotype numbers. This, and the PAT of a specific MPRK4 proportion treatment divided by the EV PAT (=PT) is used to conclude that there was an increase in yield at low proportions of MPK4. However, the summed traits could actually be measured directly. Shouldn't PAT somehow be compared to the observed traits? Was there an increase in total trait values when it was measured, rather than when it was predicted from genotype means? The authors never explain why they used PATs, or what more can be gleaned from this analyses vs. doing an analysis on measured values.

It furthermore is not clear to me if the population trends have any statistical significance. They are based on mostly non-significant differences in PATs from Figure 3, but the PT results are discussed as if they were significant. What is the statistical basis for this?

My larger criticism concerns the framing of this study, and in particular the structure of the Abstract and Introduction. For example, both discuss the importance of signals and neighbor perception as the only factors relevant for population performance. I am not debating that signals can be important in some cases, but for a manuscript that could make a contribution to the functional diversity literature this focus seems quite strange. Similarly, I found the Introduction to lack relevant references, particularly on the first page.

I highlight a few problematic passages below:

Abstract: 'Functional variation' of what? I would expect this to refer to diversity effects mediated by functional differences among species, but the paper is about trait differences in genotypes within species. Are intraspecific functional diversity effects equally well understood/reported? This statement needs to be clearer and more specific.

Abstract: It is unclear why signalling should be this important. Functional trait diversity is assumed to cause increased population yield by better niche partitioning or emergent properties of a system, which does not have to rely on plant-plant signalling. The assumption that signalling and perception would be involved in trait diversity-mediated yield effects needs to be introduced and justified better.

'this occurred at the population scale' – I have trouble imagining what this means. Could this be put more explicitly, e.g., 'while individual plants or plant pairs did not differ significantly, effects became apparent at the population scale'?

I do not understand the sentence 'higher levels in biological hierarchies set boundaries for the function of traits at lower levels in the hierarchy', and the meaning of the sentence does not become clearer from the following sentences.

Introduction, first paragraph: The hierarchical argument is weakened by the vague definitions of population and community. Do I interpret the author's definition of population correctly as 'the individuals of the same species that potentially interact within a geographical area'? Is the community then the full assemblage of all species in that same area, or does it refer to still only one species, but at a larger geographic scale? It seems the authors only ever discuss single-species communities, but this should be clearly defined.

Introduction, first paragraph: Proposed by whom?

The statement 'signals […] can originate from water-use trait variation' suggests that the signals are known and understood. However, without reading the individual cited examples, the authors appear to cite studies that show an effect of WUE variability in populations on population yield. If this is the case, this is no evidence for the involvement of signals. A signal has a clear definition, meaning the intentional transfer of information. If there is evidence for this, the authors should explain this evidence; if there is not, they should not use the term signal.

Introduction, third paragraph: I would expect a plant's yield to respond to its own soil water availability first, yet this sentence suggests that this a potential secondary mechanism, after the more important effect of sensing WUE in other plants. This seems counterintuitive.

Introduction, sixth paragraph: It seems impossible to conclude from this example that *N. attenuata* responds differently do genetically different neighbors. Perhaps the AZ plants have a faster growth or are better competitors and leave fewer resources for UT plants, in which case the result has nothing to do with UT's ability to perceive the similarity of its neighbor. A single comparison such as this one is heavily confounding genetic differences and all other differences between plants.

[Editors’ note: further revisions were suggested prior to acceptance, as described below.]

Thank you for submitting your article "Determining the scale at which variation in water-use traits and AMF associations change population yields" for consideration by *eLife*. Your article has been reviewed by 3 peer reviewers, including Bernhard Schmid as the Reviewing Editor and Reviewer #1, and the evaluation has been overseen by a Christian Hardtke as the Senior Editor. The following individuals involved in review of your submission have agreed to reveal their identity: Robert McClung (Reviewer #3).

The reviewers have discussed the reviews with one another and the Reviewing Editor has drafted this decision to help you prepare a revised submission.

Summary:

Reviewer 3 writes: "Modern agriculture relies on crops grown in monoculture. However, considerable experience and data indicate that diversity at genotypic and phenotypic levels can enhance productivity and stability. This manuscript shows that diversity associated with silencing of MPK4, enhances yield (called overyielding) in. a population in which there is a low percentage of irMPK4 plants. They authors provide data showing that the overyielding effect is not the result of interactions of near neighbors but rather emerges at a population level. Given that, the final part of the manuscript investigates communication among plants. First, they introduce genetically a second mutation, silencing *NaCCaMK*, which blocks symbiotic association with AM fungi. Nonetheless, overyielding persists, indicating that inter-plant communication does not occur via AMF connections below-ground. Finally, they perform shoot-root grafting experiments and show that the ability for a plant to respond to its neighbors requires MPK4 expression in shoots, but not in roots, strongly implicating above-ground communication in the response."

Essential revisions:

Summary of reviewer 1:

“This is a resubmission of a very challenging paper. Challenging because of its high originality and innovative methods, but also challenging because of its narrative. While the first was already impressing the reviewers of the original submission the second was so difficult initially that reviewers felt somewhat lost. This resubmitted version is now much clearer, even though the added results of mycorrhizal treatments have increased the complexity. I think with a further clarification of the narrative this will become a very important paper.

The problem with the current narrative is already visible in the title: "Determining the scale at which variation in water-use traits and AMF associations change population yields". I would change this to: "Determining the scale at which variation in a single gene changes population yields". The second title identifies the major novelty of this paper. The old title relates to a hypothesis that was carefully dismantled by the authors, namely that the reason for the single-gene diversity effect on population yield was due to its effect on water use. The story starts with two observations: 1) "overyielding" of mixtures as compared with monocultures, due to a single gene for a MAP kinase and 2) phenotypic effects of the presence vs. absence of this gene on plant water-use efficiency (WUE). This leads to the obvious hypothesis that 1) could be due to 2). However, in the end the story turns out not to support the hypothesis and the reader is left in suspense as to which other phenotypic effects of the gene could be responsible for 1).

There are some indications in the manuscript of how the suspense could be reduced, but the main revisions should be aimed at improving this aspect of the paper (including a better final sentence in the Abstract). First, it is already mentioned in the Introduction that a gene that affects MAPKs can have multiple phenotypes, changed WUE only being one of those and here thoroughly tested and excluded as cause of overyielding. Second, in the Discussion it is mentioned that other factors causing overyielding could range from niche complementarity to the exchange of chemical signals. But the discussion then still comes back to below-ground mechanisms whereas I think it would be better to focus on potential above-ground mechanisms and a more in-depth "speculation" of how these could work.”

Reviewer 2 finds the second part of the paper difficult to follow, because one could look in different ways at the difference between the performance of a plant grown in a pair vs. alone. This requires careful explanation, as suggested by this reviewer:

"I find the revised manuscript by McGale et al. has been improved considerably, and both the new Introduction and figures make this complex story more approachable. In particular the first half of experiments are very well set up conceptually now, and the sequence of results makes intuitive sense. Unfortunately, I find this is not maintained throughout the manuscript, and for some of the later experiments I do not follow the conclusions that the authors draw from them.

The central observation of the study is that in populations of empty-vector (EV) control plants and low-water use efficiency (irMPK4) plants, a low proportion of irMPK4 plants causes population overyielding. The authors then carry out a series of experiments to identify potential mechanisms explaining this overyielding. Through population experiments in the field and greenhouse, they demonstrate that overyielding is caused by an increased performance of EV plants, apparently at no cost to irMPK4 plants. While this increase is not significant at the individual plant level it seems nonetheless fairly apparent in Figure 3—figure supplement 2, and the cumulative effect on all EV plants results in a significant population increase, demonstrated in Figure 3. Interestingly, the authors did not find an effect of overyielding when EV plants were grown with a single irMPK4 neighbour, leading the authors to conclude that overyielding is a population-scale process.

However, there seems to be a logical disconnect between the population results on overyielding, and the experiments evaluating MPK4 in the paired design. In the paired design, the authors demonstrate that irMPK4 plants all have the same (lower) biomass and fitness correlates, regardless of whether they are grown individually or in pairs with EV or irMPK4 plants, while EV plants have better performance when growing alone. The authors conclude that this means MPK4 is required for the *N. attenuata* growth and yield response (subsection “MPK4 is necessary for *N. attenuata’s* growth and yield responses to neighbors”).

I have two issues with this conclusion. First, it seems to me this conclusion does not match the data: I would think if MPK4 were involved in a growth/yield response to neighbours, the irMPK4 plants should all have the same high biomass/yield of EV plants grown alone. Instead, the data suggests to me that irMPK4 plants are impaired in growth, making them unable to benefit from reduced competition when growing alone.

Second, I find it difficult to connect these results and those of the grafting experiment to the observed overyielding. The results demonstrate that MPK4 expressed in shoot allows plant to benefit from growing alone, while abrogation of MPK4 in the shoot 'impairs' plant fitness and reduces it to the level of competing plants. However, the authors clearly demonstrate that overyielding was achieved by plants increasing growth/yield while surrounded by many neighbours – the MPK4 effect on plants growing alone thus does not seem relevant for explaining overyielding.

I assume these issues could simply be resolved by a more careful transition between the two experimental parts, and by a more extensive explanation of the deductive reasoning used here. I would think this may benefit many of the future readers of this study."

Reviewer 3 has general points that are also indicated in the summary above:

"I have two major concerns:

1) The title is misleading because the AMF associations or lack thereof did not affect population yields. Second, I think it may be premature to argue that the WUE phenotypic consequences of irMPK4 are responsible-see next point.

2) irMPK4 results in reduced water use efficiency (WUE), but it is not at all clear that the overyielding observed here is associated with the WUE aspects of loss of MPK4. Indeed, do not the rewatering experiments in which they add to each pot the amount of water predicted to be lost each day not argue against (or at least fail to support) a role for WUE? The authors argue that altered WUE may be responsible because the rate of water loss from the soil around the irMPK4 plants will be more rapid than around wild type MPK4+ plants, which is certainly true. It is equally true that many changes may be associated with very early stages of drought, probably like those encountered by the irMPK4 plants in these experiments. However, it is important to remember that MAP kinase cascades can signal through multiple disparate downstream targets to influence a broad array of phenotypes. The effect of irMPK4 in overyielding may be unrelated to its effect on WUE."

---

## [Author Response]

[Editors’ note: the authors resubmitted a revised version of the paper for consideration. What follows is the authors’ response to the first round of review.]

The strength of you manuscript is the use of very promising tools to analyze a within-species diversity effect mechanistically down to the effects of a single genetic difference between members of a population. The weakness is the ecological setting of the work as detailed in the reviews below. This aspect could be improved by setting a different frame, but we also feel it would need different or additional experiments.

We agree with these concerns and have addressed the two weaknesses as follows: first, the data has now been visualized and contextualized in consideration to the ecological literature as suggested by the reviewers and the Editor. Second, we include additional data pertaining to our analysis of the influence of arbuscular mycorrhizal fungi (AMF) networks on the yields of our field populations (Figure 5C; Figure 7). We think that this adds to the strength of the manuscript as an additional tool to identify the scale at which trait diversity causes population overyielding.

All reviewers found your manuscript very difficult to understand, and I think some of the comments reflect this, for example those about statistics in the third review.

We have substantially improved the visualization of our overyielding results by using de Wit diagrams (suggested by reviewer #1; applied in Figures 3, 4, 7). This significantly reduced the number of new abbreviations that were needed and allowed us to rely on established terminology. We acknowledge that the descriptions of the statistical analyses were not sufficient and have added additional information in our statistical tables as well as in our Materials and methods section. We hope that both these changes help to clarify our results and statistics, and that the rigor of the work can be more readily appreciated.

Reviewer #1:[…] Mechanisms of how emergent features can be created by diversity have been discussed at large in the biodiversity-ecosystem functioning literature, but their molecular basis has almost never been assessed, except in a recent paper by Wuest and Niklaus (Nature Ecology and Evolution 2018).

We found this remark very insightful and included this and similar references in the first paragraph of our Introduction. Particularly, we found that there have been several instances where forward genetics methods have shown a particular locus or chromosomal region as potentially driving observed biodiversity-productivity relationships and that they have not been tested further. This may be due to the complexity of population experiments and uncertainty regarding the scale at which to test the function of these loci and observe their potential effects. We believe that citing this literature has helped to highlight the need for work determining the effects of

certain gene-trait alterations at different scales within populations.

The authors focus in my view perhaps too much on a comparison between neighbor interactions and effects at neighborhood scale and population-level responses. This originates from the hypothesis that it could be chemical signals that are exchanged between individuals that lead to the observed phenomena. However, other, perhaps more phenomenological mechanisms would be conceivable and would typically be considered in biodiversity-ecosystem functioning research. These include resource complementarity or interactions with other organisms, including soil microbes. Basically, the question is why a few individuals that have lost the ability to control their stomates can increase population performance.

We considered this point carefully in light of the observation that we interpreted the results, specifically in relation to chemical signals, without discussing alternative hypotheses and have removed this unwarranted early emphasis on chemical signals (Figure 1; Abstract, Introduction, Discussion). Instead, we use new experimental data from populations connected to AMF networks to continue our line of inquiry in identifying the scale at which overyielding responses may occur. We emphasize in the Discussion that this work is essential to determine mechanism through which MPK4 acts at the above-ground population scale to result in population reproductive overyielding, whether this can be attributed to specific chemical signals, or to other plant-plant interaction phenomena such as complementation.

Reviewer #2:[…] 1) Water availability: studying the functional implications of the presence of low-WUE individuals calls for experiments in which water availability is deliberately varied, controlled and monitored. Such a design would contrast high-water conditions where the presence of low-WUE individuals would be expected to cause little population-level effects and low-water conditions under which the presence of such plants would be expected to trigger significant population-level effects. While the authors tried to create differential water treatments in the field experiment, this attempt did not succeed. Moreover, in the two greenhouse experiments differential water availability was not even attempted and instead the authors provided constant water supply by applying frequent water supplements.

We agree that controlling for soil water availability is the best means of studying the functional implications of the presence of low WUE individuals in populations, however, we disagree that this requires contrasting conditions of low- to high-water availability, especially if these two conditions do not represent alternative selective regimes for plants in nature. The field results demonstrate that soil moisture in the natural habitat of *N. attenuata* decreases throughout the field season (Figure 3—figure supplement 4), where high water availability is only relevant at earlier developmental stages of the plant and low water availability at later stages. This is commonly the case in the Great Basin Desert where the majority of the annual precipitation falls in the winter, before the start of the growing season. This decrease did not appear to vary by microenvironment within our populations and did not depend on a period of withheld irrigation (Figure 3—figure supplement 4). In our glasshouse experiments, we considered the ecological importance of the changes in water availability across developmental stages, and now include a description in Materials and methods of our strategies to vary glasshouse water availability in an ecologically relevant manner. In short, we began experiments with a period of high soil moisture for establishment, transitioned to soil moisture percentages that were similar to those in the field for the developmental transitions into bolting and anthesis, and then withheld water for natural and equal dry-downs across pots in the glasshouse at the end of each experiment. This regime of controlled decreases in water availability transitions the populations into producing ripe seed capsules due to the scarcity of soil water, mimicking the natural drydown that occurs in nature, which stimulates capsule ripening regardless of plant age after anthesis (Zavala and Baldwin, 2004 and references within). A dry-down through our controlled watering technique is visually demonstrated in Figure 4 using our paired experiment to emphasize the use of this method that ensures that all pots synchronously reach 0g of available water. We describe this method in greater detail in our unpublished results (McGale and Valim et al) in which the benefits of this approach in attributing plant responses to equal changes in water variability are highlighted.

Besides missing the above-mentioned aspect of differential water availability, this protocol might or might not have sufficiently homogenized water availability for the plants. Even if soil water content seemed to have been equalized by the daily supplementation, soil and plants surely underwent daily cycles of diurnal drying (more so in populations with larger proportions of low WUE plants), which could have had important effects on plant drought perception, responses and population effects. Overcoming this problem could have been achieved by running at least two (high vs. low) water treatments that would be continuously sustained with high precision and temporal resolution (minutes, not whole days) and soil water content should have been monitored using non-destructive methods, e.g. TDR, to account for high-res variability in water availability.

We homogenized water availability for the plants (see above response and our additional reference), and did not find that different cycles of diurnal drying or potential drought perception or responses occurred between the two genotypes in differing population types. This was demonstrated through the use of Yara- Zim leaf turgor pressure probes, which were attached to the leaves of EV and irMPK4 plants in the populations and provided continuous recordings throughout the experiment. We now include these data in Figure 3—figure supplement 8 and a detailed interpretation of the graphs in our Materials and methods section. These data from individual plants are consistent with the inferences from the controlled water experiments.

Accordingly, while the findings of these experiments are legitimate, their interpretation could be different. Specifically, it can be hypothesized that the observed population-level responses to the presence of low proportions of low-WUE plants was triggered by the slightly reduced water availability in those treatments (rendered by these plants) and not necessarily by any non-resource signaling or cuing. Teasing apart the resource availability and non-resource information alternatives would require additional treatments and much tighter monitoring of soil water (see above).

In the glasshouse experiments, we found that the grams of water used by a particular pot could be linearly predicted; equal soil water availability was consistently achieved across all pots after a 24-hour drying period (e.g. troughs in Figure 4B) when pots were provided with exactly the amount that they would use in one day (e.g. peaks in Figure 4B). Pots which used more water (i.e. with more irMPK4 plants) received more grams of water at watering events (peaks in Figure 4B). From these greater amounts of water, pots with a greater proportion of irMPK4 plants would have a steeper decline in water availability throughout the day. This would mean that populations with higher frequency of low-WUE plants would surpass low frequency low-WUE plant populations in depleting water availability throughout the day and therefore they would have equal- or lower-water availability for a longer period. This could preferentially trigger growth responses in those populations, as stated by reviewer #2. We did not observe overyielding in these high frequency populations, nor did we observe unequal water availability in any pot type at the end of each 24-hour cycle.

2) Neighbor- vs. population-level effects: based on careful analyses of the response of individuals located at variable distances from the irMPK4 plants, the authors claim that the population-level effects of the presence of low proportion irMPK4 plants could not be attributed to local interactions between immediate neighbors; however, given that a) the effect is most likely done via root-root interactions and b) the precise position of the roots in such micro populations is not known, the reported results could be hypothesized to have been caused by the very commonly observed aggregation of more active roots near the pot's boundary/wall, where most root interaction took place. Accordingly, the seemingly counter-intuitive 'scale-skipping' phenomenon could be trivially attributed to a technical artifact rather than to an ecologically meaningful phenomenon.

The experiments with grafted plants having depleted MPK4 expression only in roots, which responded in a manner similar to that of the EV control plants, indicated that shoot MPK4 expression, not root expression, is responsible for the phenotypes attributed to irMPK4. These data do not eliminate the possibility of root interactions contributing to the phenomena that we observe but do indicate that shoot phenotypes and above-ground interactions are more important for the differences introduced by MPK4 silencing. The new data presented in Figure 7 and accompanying conclusions are consistent with the grafting results and have redirected the discussion to considerations of above-ground interactions rather than below-ground phenomena in the field or the glasshouse.

We removed the discussion of pot-side effects as we could not find strong references to support that these would significantly change the interactions of plants in population (see response to reviewer #1). Additionally, the consistency of the overyielding in the glasshouse with that in the field, without pot side effects and with fewer direct neighbor interactions, lead us to infer that direct root contact influenced by pot-side interactions is not likely the main driver of the observed overyielding.

3) Abrogation of AMF associations: the hypothesis that mycorrhizal networks could play an important functional role in the priming of neighboring plants to various stresses has been supported in other cases and could have been interesting in the context of the present project. It is thus not clear why the authors chose to eliminate this aspect from the study.

We agreed strongly with this point and now include this dataset in a new figure (Figure 7).

4) Fitness effects: while proxies of fitness typically, and for very good reasons, serve as golden evidence in ecological studies, attention should be given to the fact that the harvests and fitness surveys were conducted at ca. day 50 and before plants had the opportunity to finish their natural life cycle (typically 90 d). Given that the presence of a low proportion of low-WUE plants could have slightly reduced water availability, the observed differences could be attributed to accelerated reproduction, which is rather common in plans responding mild stresses or signals and cues of imminent stresses (see e.g. https://doi.org/10.1093/jxb/erw272).

We hope we have addressed both of these points in our previous responses to the important water availability considerations raised by reviewer #2.

5) Shoot involvement in MPK4-deficiency: the demonstration of the importance of the shoots in mediating the functionality of MPK4-deficiency is very nice but not directly related to the main questions dealt with in the study. Given the vastness of the manuscript, it seems that this finding could be reported elsewhere.

We agree that this previously appeared tangential to the main points of the manuscript. However, this dataset demonstrating that mediation of growth responses to neighbors by MPK4 is shoot-localized is consistent with the argument pointing to the importance of above-ground interactions in the phenomena we observe. We hope that this has become clearer with the inclusion of data from AMF-associated populations.

6) Data analyses and presentation: a) Analysis of a large proportion of the results could have been better done using ANCOVAs, accounting for individual data, not merely treatment means.

Analysis of the data was done using ANOVAs as the assumptions of ANCOVA would have been violated: this has now been clarified in the Materials and methods sections. We acknowledge that this may not have been clear due to our use of different terminology

associated with the analyses.

b) The graphs are extremely cumbersome, complex and hard to follow. Lumping result by analysed plant and not paired by treatment is extremely hard to follow and non-intuitive. The tiny trend insets seem jargony as the order of the presented data is not intuitive (see previous comment).c) Why CIs are not presented for population-level data?e) In most cases, the presentation of means could be much clearer using colored points and CI bars rather than colored bars.

We hope that the conversion of the data into de Wit diagrams and the inclusion of means and error structures in new supplementary figures (see response to reviewer #1) addresses both of these concerns.

d) The main text is heavy with statistical information that could be moved to tables or to figure legends.

We thank the reviewer for this feedback, as it helped us to improve the clarity and continuity of the manuscript. We addressed this by moving statistical information from the main text into 3 new main data tables. We additionally corrected the mistakes indicated in the additional points provided by reviewer #2.

Reviewer #3:McGale et al. evaluate the role of a single gene, MPK4, on the population yield of *Nicotiana attenuata*. In general, the experiments in the study appear to have been well-designed and comprise an elegant set of tests in the search for a mechanistic explanation of the increase in yield.Regarding the different methods, I did not follow the author's justification for using PATs. The authors calculate mean traits for each genotype in each treatment, and then calculate the predicted summed traits from genotype means and genotype numbers. This, and the PAT of a specific MPRK4 proportion treatment divided by the EV PAT (=PT) is used to conclude that there was an increase in yield at low proportions of MPK4. However, the summed traits could actually be measured directly. Shouldn't PAT somehow be compared to the observed traits? Was there an increase in total trait values when it was measured, rather than when it was predicted from genotype means? The authors never explain why they used PATs, or what more can be gleaned from this analyses vs. doing an analysis on measured values.It furthermore is not clear to me if the population trends have any statistical significance. They are based on mostly non-significant differences in PATs from Figure 3, but the PT results are discussed as if they were significant. What is the statistical basis for this?

We hope that our implementation of de Wit diagrams (see responses to reviewer #1, #2) to demonstrate overyielding in the field and glasshouse helped to clarify the presentation of the results which were previously shown using PATs and PTs. We included means and error structures for the data that was used to create the de Wit diagrams in Figures 3—figure supplement 2 and Figure 3—figure supplement 6.

My larger criticism concerns the framing of this study, and in particular the structure of the Abstract and Introduction. For example, both discuss the importance of signals and neighbor perception as the only factors relevant for population performance. I am not debating that signals can be important in some cases, but for a manuscript that could make a contribution to the functional diversity literature this focus seems quite strange. Similarly, I found the Introduction to lack relevant references, particularly on the first page.

We thank reviewer #3 for these comments and for the highlighted passages included in the review. We re-wrote the Introduction to address these comments, and also removed the emphasis on chemical signals, rather than other ecological mechanisms that could have produced our results (see response to reviewer #1). We introduce and discuss a variety of mechanisms that could contribute to the observed overyielding effects. Additionally, we focused the novelty of our manuscript at determining the scale at which these effects occur and on setting a foundation for exploring a variety of discussed mechanisms (see response to reviewer #1). We included many additional references, particularly in the Introduction, and hope that this helps to situate the work appropriately in the context of functional diversity literature.

[Editors’ note: what follows is the authors’ response to the second round of review.]

Essential revisions:Summary of reviewer 1:“This is a resubmission of a very challenging paper. Challenging because of its high originality and innovative methods, but also challenging because of its narrative. While the first was already impressing the reviewers of the original submission the second was so difficult initially that reviewers felt somewhat lost. This resubmitted version is now much clearer, even though the added results of mycorrhizal treatments have increased the complexity. I think with a further clarification of the narrative this will become a very important paper.The problem with the current narrative is already visible in the title: "Determining the scale at which variation in water-use traits and AMF associations change population yields". I would change this to: "Determining the scale at which variation in a single gene changes population yields". The second title identifies the major novelty of this paper. The old title relates to a hypothesis that was carefully dismantled by the authors, namely that the reason for the single-gene diversity effect on population yield was due to its effect on water use.

We agree with reviewers 1 and 3 (please see reviewer #3 comments 1 and 2) regarding the misleading nature of our previous title and the demonstration of WUE- and AMF-independent population yield effects. We have addressed this directly by changing the title to "Determining the scale at which variation in a single gene changes population yields"in order for it to accurately reflect the content of the paper.

The story starts with two observations: 1) "overyielding" of mixtures as compared with monocultures, due to a single gene for a MAP kinase and 2) phenotypic effects of the presence vs. absence of this gene on plant water-use efficiency (WUE). This leads to the obvious hypothesis that 1) could be due to 2). However, in the end the story turns out not to support the hypothesis and the reader is left in suspense as to which other phenotypic effects of the gene could be responsible for 1).There are some indications in the manuscript of how the suspense could be reduced, but the main revisions should be aimed at improving this aspect of the paper (including a better final sentence in the Abstract).

In accordance with the title change, we agree that the Abstract needed to be revised in order to accurately reflect the progressive dissection of the overyielding effect through the experimental narrative of the manuscript. We have changed several sentences, including the last one, in hope of clarifying the sequential experimental steps in the dissection of the source of the observed overyielding response: first, that the effect results from the single-gene manipulation, and not from water-use phenotypes; second, that it occurs at the population scale; and third, that it likely specifically occurs above-ground.

First, it is already mentioned in the Introduction that a gene that affects MAPKs can have multiple phenotypes, changed WUE only being one of those and here thoroughly tested and excluded as cause of overyielding. Second, in the Discussion it is mentioned that other factors causing overyielding could range from niche complementarity to the exchange of chemical signals. But the discussion then still comes back to below-ground mechanisms whereas I think it would be better to focus on potential above-ground mechanisms and a more in-depth "speculation" of how these could work.”

Incorporating this advice from reviewer #1 was valuable to improve the thoroughness of our Discussion. We have now included more in-depth speculations on the extended mechanisms that could manifest in neighbor- versus population-scale effects (Discussion, third paragraph, as well as bolstered our speculation on potential above-ground mechanisms, including adding several new paragraphs (Discussion, seventh to tenth paragraphs).

Reviewer 2 finds the second part of the paper difficult to follow, because one could look in different ways at the difference between the performance of a plant grown in a pair vs. alone. This requires careful explanation, as suggested by this reviewer:"I find the revised manuscript by McGale et al. has been improved considerably, and both the new Introduction and figures make this complex story more approachable. In particular the first half of experiments are very well set up conceptually now, and the sequence of results makes intuitive sense. Unfortunately, I find this is not maintained throughout the manuscript, and for some of the later experiments I do not follow the conclusions that the authors draw from them.The central observation of the study is that in populations of empty-vector (EV) control plants and low-water use efficiency (irMPK4) plants, a low proportion of irMPK4 plants causes population overyielding. The authors then carry out a series of experiments to identify potential mechanisms explaining this overyielding. Through population experiments in the field and greenhouse, they demonstrate that overyielding is caused by an increased performance of EV plants, apparently at no cost to irMPK4 plants. While this increase is not significant at the individual plant level it seems nonetheless fairly apparent in Figure 3—figure supplement 2, and the cumulative effect on all EV plants results in a significant population increase, demonstrated in Figure 3. Interestingly, the authors did not find an effect of overyielding when EV plants were grown with a single irMPK4 neighbour, leading the authors to conclude that overyielding is a population-scale process.However, there seems to be a logical disconnect between the population results on overyielding, and the experiments evaluating MPK4 in the paired design. In the paired design, the authors demonstrate that irMPK4 plants all have the same (lower) biomass and fitness correlates, regardless of whether they are grown individually or in pairs with EV or irMPK4 plants, while EV plants have better performance when growing alone. The authors conclude that this means MPK4 is required for the *N. attenuata* growth and yield response (subsection “MPK4 is necessary for *N. attenuata*’s growth and yield responses to neighbors”).I have two issues with this conclusion. First, it seems to me this conclusion does not match the data: I would think if MPK4 were involved in a growth/yield response to neighbours, the irMPK4 plants should all have the same high biomass/yield of EV plants grown alone. Instead, the data suggests to me that irMPK4 plants are impaired in growth, making them unable to benefit from reduced competition when growing alone.Second, I find it difficult to connect these results and those of the grafting experiment to the observed overyielding. The results demonstrate that MPK4 expressed in shoot allows plant to benefit from growing alone, while abrogation of MPK4 in the shoot 'impairs' plant fitness and reduces it to the level of competing plants. However, the authors clearly demonstrate that overyielding was achieved by plants increasing growth/yield while surrounded by many neighbours – the MPK4 effect on plants growing alone thus does not seem relevant for explaining overyielding.I assume these issues could simply be resolved by a more careful transition between the two experimental parts, and by a more extensive explanation of the deductive reasoning used here. I would think this may benefit many of the future readers of this study."

We found these suggestions from reviewer 2 essential in improving the logical continuity of the manuscript. We extensively edited the Discussion in order to distinguish between the two main methods that we used: first, comparisons among individuals across population types; and second, analyses of the neighbor responses of individual genotypes in order to infer (although, not test directly) how these might potentially result in emergent overyielding effects at the population scale. The latter was tested using comparisons of individuals planted alone or in pairs, versus individuals planted in varying population types. The observations of reviewer 2 are completely correct: the data at first seems contradictory, as EV plants in single and paired plantings (i.e. in the absence or presence of a neighbor) reveal that EV reduces yield in the presence of a neighbor (EV or irMPK4), but in populations with varying irMPK4 percentages, overyielding occurs in comparison to EV monocultures. However, these results emerge from two different methods that do not necessarily produce results that are mutually exclusive. The former result includes a density test that is not present in experiments comparing individuals across pairs or populations with a consistent number of individuals. It is entirely possible that in our 12-plant monocultures, EV has reduced yield in comparison to a single plant in the 7.5L population pots. EV’s density-related yield reduction (i.e. with increasing numbers of neighbors) may be attenuated by different interaction scenarios with irMPK4 plants (i.e. with increasing neighbor “quality”), leading to overyielding in comparison to EV in monocultures, but not necessarily in comparison to a single EV. We use comparisons with the single EV in a subset of our paired experiments to evaluate the neighbor response profiles of EV and irMPK4 in order to nucleate our speculations about the possible mechanisms responsible for the overyielding, but we fully agree that direct comparisons are not warranted. We see now that our previous descriptions implied that these neighbor response profiling experiments were directly explaining overyielding, and have carefully corrected these statements. We have also recognized that the transitions between the two methods were at best, telegraphic, in our previous Discussion, so this section has been embellished.

Reviewer 2 brings to light a good point regarding our discussion of *MPK4* as a requirement for *N. attenuata’s* response to neighbors. Our tests, as well as previous studies, show that irMPK4 grows smaller than EV. In our previous discussion, we inferred that the lack of a difference in irMPK4’s growth with or without a neighbor demonstrates that irMPK4 does not respond to neighbors. In fact, we are not able to distinguish between a lack of response to neighbors and an impaired growth phenotype that does not allow irMPK4 (or shoot-*MPK4*-deficient) plants to benefit from the absence of a neighbor, as stated by reviewer 2. We have addressed this in our discussion by using more precise language to what we have observed: irMPK4 plants’ growth and reproductive yield does not change in the presence of a neighbor, while EV plants’ does. Although we cannot specifically dissect the mechanism of this differential response, we still believe it to be a significant finding from which we can infer (but not directly test) how *MPK4* plays a role in *N. attenuata’s* neighbor responses. We have made the description and discussion of this result more precise throughout the manuscript, and have congruently elaborated on our related inferences in the Discussion.

We also agree that the purpose of our paired grafting experiment was not well motivated in the Discussion. This experiment is now motivated as an exploration of the above- or below-ground spatial scales at which population-scale effects may have occurred. The paired grafting experiment was a neighbor absence or presence test with the intent to identify the tissues specifically involved in *N. attenuata’s* ability to respond to a neighbor. Reviewer 2 noted correctly that our previous Discussion implied that the grafted pair results directly explained overyielding results. This has been corrected and overall we have attempted to better align results with inferences. We believe that these changes clarify the main novel findings of the manuscript, and point to potential future steps that could directly explore the relationship between *MPK4* abundanceand population overyielding.

Reviewer 3 has general points that are also indicated in the summary above:"I have two major concerns:1) The title is misleading because the AMF associations or lack thereof did not affect population yields. Second, I think it may be premature to argue that the WUE phenotypic consequences of irMPK4 are responsible-see next point.2) irMPK4 results in reduced water use efficiency (WUE), but it is not at all clear that the overyielding observed here is associated with the WUE aspects of loss of MPK4. Indeed, do not the rewatering experiments in which they add to each pot the amount of water predicted to be lost each day not argue against (or at least fail to support) a role for WUE? The authors argue that altered WUE may be responsible because the rate of water loss from the soil around the irMPK4 plants will be more rapid than around wild type MPK4+ plants, which is certainly true. It is equally true that many changes may be associated with very early stages of drought, probably like those encountered by the irMPK4 plants in these experiments. However, it is important to remember that MAP kinase cascades can signal through multiple disparate downstream targets to influence a broad array of phenotypes. The effect of irMPK4 in overyielding may be unrelated to its effect on WUE."

Please see our first response to Essential revisions above.